# Leverage the Average: an Analysis of KL Regularization in Reinforcement Learning

**Nino Vieillard**
Google Research, Brain Team
Université de Lorraine, CNRS, Inria
IECL, F-54000 Nancy, France
vieillard@google.com

**Tadashi Kozuno**[*]
Okinawa Institute of Science and Technology
tadashi.kozuno@oist.jp

**Bruno Scherrer**
Université de Lorraine, CNRS, Inria
IECL, F-54000 Nancy, France
bruno.scherrer@inria.fr

**Olivier Pietquin**
Google Research, Brain Team
pietquin@google.com

**Rémi Munos**
DeepMind
munos@google.com

**Matthieu Geist**
Google Research, Brain Team
mfgeist@google.com

## Abstract

Recent Reinforcement Learning (RL) algorithms making use of Kullback-Leibler (KL) regularization as a core component have shown outstanding performance. Yet, only little is understood theoretically about why KL regularization helps, so far. We study KL regularization within an approximate value iteration scheme and show that it implicitly averages $q$-values. Leveraging this insight, we provide a very strong performance bound, the very first to combine two desirable aspects: a linear dependency to the horizon (instead of quadratic) and an error propagation term involving an averaging effect of the estimation errors (instead of an accumulation effect). We also study the more general case of an additional entropy regularizer. The resulting abstract scheme encompasses many existing RL algorithms. Some of our assumptions do not hold with neural networks, so we complement this theoretical analysis with an extensive empirical study.

## 1 Introduction

In Reinforcement Learning (RL), Kullback-Leibler (KL) regularization consists in penalizing a new policy from being too far from the previous one, as measured by the KL divergence. It is at the core of efficient deep RL algorithms, such as Trust Region Policy Optimization (TRPO) [37] (motivated by trust region constraints) or Maximum a Posteriori Policy Optimization (MPO) [2] (arising from the view of control as probabilistic inference [26, 16]), but without much theoretical guarantees. Recently, Geist et al. [20] have analyzed algorithms operating in the larger scope of regularization by Bregman divergences. They concluded that regularization doesn't harm in terms of convergence, rate of convergence, and propagation of errors, but these results are not better than the corresponding ones in unregularized approximate dynamic programming (ADP).

---

[*]Work done while at DeepMind.

Building upon their formalism, we show that using a KL regularization implicitly averages the successive estimates of the $q$-function in the ADP scheme. Leveraging this insight, we provide a strong performance bound, the very first to combine two desirable aspects: 1) it has a linear dependency to the time horizon $(1-\gamma)^{-1}$, 2) it exhibits an error averaging property of the KL regularization. The linear dependency in the time horizon contrasts with the standard quadratic dependency of usual ADP, which is tight [35]. The only approaches achieving a linear dependency we are aware of make use of non-stationary policies [8, 35] and never led to practical deep RL algorithms. More importantly, the bound involves the norm of the average of the errors, instead of a discounted sum of the norms of the errors for classic ADP. This means that, while standard ADP is not guaranteed to converge for the ideal case of independent and centered errors, KL regularization allows convergence to the optimal policy in that case. The sole algorithms that also enjoy this compensation of errors are Dynamic Policy Programming (DPP) [7] and Speedy Q-learning (SQL) [6], that also build (implicitly) on KL regularization, as we will show for SQL. However, their dependency to the horizon is quadratic, and they are not well amenable to a deep learning setting [43].

We also study the case of an additional entropy regularization, usual in practical algorithms, and specifically the interplay between both regularizations. The resulting abstract framework encompasses a wide variety of existing RL algorithms, the connections between some of them being known [20], but many other being new, thanks to the implicit average of $q$-values. We highlight that, even though our analysis covers the case where only the entropy regularization is considered, it does not explain why it helps without an additional KL term. Some argue that having a higher entropy helps exploration [38], other that it has beneficial effects on the optimization landscape [3], but it also biases the solution of the MDP [20].

Our analysis requires some assumptions, notably that the regularized greedy step is done without approximation. If this is reasonable with discrete actions and a linear parameterization, it does not hold when neural networks are considered. Given their prevalence today, we complement our thorough analysis with an extensive empirical study, that aims at observing the core effect of regularization in a realistic deep RL setting.

## 2 Background and Notations

Let $\Delta_X$ be the set of probability distributions over a finite set $X$ and $Y^X$ the set of applications from $X$ to the set $Y$. An MDP is a tuple $\{\mathcal{S}, \mathcal{A}, P, r, \gamma\}$ with $\mathcal{S}$ the finite state space, $\mathcal{A}$ the finite set of actions, $P \in \Delta_{\mathcal{S}}^{\mathcal{S} \times \mathcal{A}}$ the Markovian transition kernel, $r \in \mathbb{R}^{\mathcal{S} \times \mathcal{A}}$ the reward function bounded by $r_{\max}$, and $\gamma \in (0,1)$ the discount factor. For $\tau \geq 0$, we write $v_{\max}^\tau = \frac{r_{\max} + \tau \ln |\mathcal{A}|}{1-\gamma}$ and simply $v_{\max} = v_{\max}^0$. We write $\mathbf{1} \in \mathbb{R}^{\mathcal{S} \times \mathcal{A}}$ the vector whose components are all equal to 1. A policy $\pi \in \Delta_{\mathcal{A}}^{\mathcal{S}}$ associates a distribution over actions to each state. Its (state-action) value function is defined as $q_\pi(s,a) = \mathbb{E}_\pi \left[ \sum_{t=0}^\infty \gamma^t r(S_t, A_t) | S_0 = s, A_0 = a \right]$, $\mathbb{E}_\pi$ being the expectation over trajectories induced by $\pi$. Any optimal policy satisfies $\pi_* \in \operatorname{argmax}_{\pi \in \Delta_{\mathcal{A}}^{\mathcal{S}}} q_\pi$ (all scalar operators applied on vectors should be understood point-wise), and $q_* = q_{\pi_*}$. The following notations will be useful. For $f_1, f_2 \in \mathbb{R}^{\mathcal{S} \times \mathcal{A}}, \langle f_1, f_2 \rangle = \left( \sum_a f_1(s,a) f_2(s,a) \right)_s \in \mathbb{R}^{\mathcal{S}}$. This will be used with $q$-values and (log) policies. We write $P_\pi$ the stochastic kernel induced by $\pi$, and for $q \in \mathbb{R}^{\mathcal{S} \times \mathcal{A}}$ we have $P_\pi q = \left( \sum_{s'} P(s'|s,a) \sum_{a'} \pi(a'|s') q(s',a') \right)_{s,a} \in \mathbb{R}^{\mathcal{S} \times \mathcal{A}}$. For $v \in \mathbb{R}^{\mathcal{S}}$, we also define $Pv = \left( \sum_{s'} P(s'|s,a) v(s') \right)_{s,a} \in \mathbb{R}^{\mathcal{S} \times \mathcal{A}}$, hence $P_\pi q = P \langle \pi, q \rangle$.

The Bellman evaluation operator is $T_\pi q = r + \gamma P_\pi q$, its unique fixed point being $q_\pi$. The set of greedy policies w.r.t. $q \in \mathbb{R}^{\mathcal{S} \times \mathcal{A}}$ is $\mathcal{G}(q) = \operatorname{argmax}_{\pi \in \Delta_{\mathcal{A}}^{\mathcal{S}}} \langle q, \pi \rangle$. A classical approach to estimate an optimal policy is Approximate Modified Policy Iteration (AMPI) [34, 36],

$$\begin{cases} \pi_{k+1} \in \mathcal{G}(q_k) \\ q_{k+1} = (T_{\pi_{k+1}})^m q_k + \epsilon_{k+1} \end{cases},$$

which reduces to Approximate Value Iteration (AVI, $m = 1$) and Approximate Policy Iteration (API, $m = \infty$) as special cases. The term $\epsilon_{k+1}$ accounts for errors made when applying the Bellman operator. For example, the classic DQN [27] is encompassed by this abstract ADP scheme, with $m = 1$ and the error arising from fitting the neural network

(regression step of DQN). The typical use of $m$-step rollouts in (deep) RL actually corresponds to an AMPI scheme with $m > 1$. Next, we add regularization to this scheme.

## 3   Regularized MPI

In this work, we consider the entropy $\mathcal{H}(\pi) = -\langle \pi, \ln \pi \rangle \in \mathbb{R}^{\mathcal{S}}$ and the KL divergence $\mathrm{KL}(\pi_1 \| \pi_2) = \langle \pi_1, \ln \pi_1 - \ln \pi_2 \rangle \in \mathbb{R}^{\mathcal{S}}$. First, we introduce a slight variation of the Mirror Descent MPI scheme [20] (handling both KL and entropy penalties, based on $q$-values).

**Mirror Descent MPI.**   For $q \in \mathbb{R}^{\mathcal{S} \times \mathcal{A}}$ and an associated policy $\mu \in \Delta_{\mathcal{A}}^{\mathcal{S}}$, we define the regularized greedy policy as $\mathcal{G}_{\mu}^{\lambda,\tau}(q) = \operatorname{argmax}_{\pi \in \Delta_{\mathcal{A}}^{\mathcal{S}}}(\langle \pi, q \rangle - \lambda \operatorname{KL}(\pi \| \mu) + \tau \mathcal{H}(\pi))$. Observe that with $\lambda = \tau = 0$, we get the usual greediness. Notice also that with $\lambda = 0$, the KL term disappears, so does the dependency to $\mu$. In this case we write $\mathcal{G}^{0,\tau}$. We also account for the regularization in the Bellman evaluation operator. Recall that the standard operator is $T_{\pi}q = r + \gamma P \langle \pi, q \rangle$. Given the form of the regularized greediness, it is natural to replace the term $\langle \pi, q \rangle$ by the regularized one, giving $T_{\pi|\mu}^{\lambda,\tau}q = r + \gamma P (\langle \pi, q \rangle - \lambda \operatorname{KL}(\pi \| \mu) + \tau \mathcal{H}(\pi))$. These lead to the following MD-MPI($\lambda,\tau$) scheme. It is initialized with $q_0 \in \mathbb{R}^{\mathcal{S} \times \mathcal{A}}$ such that $\|q_0\|_{\infty} \leq v_{\max}$ and with $\pi_0$ the uniform policy, without much loss of generality (notice that the greedy policy is unique whenever $\lambda > 0$ or $\tau > 0$):

$$\begin{cases} \pi_{k+1} = \mathcal{G}_{\pi_k}^{\lambda,\tau}(q_k) \\ q_{k+1} = (T_{\pi_{k+1}|\pi_k}^{\lambda,\tau})^m q_k + \epsilon_{k+1} \end{cases} . \tag{1}$$

**Dual Averaging MPI.**   We provide an equivalent formulation of scheme (1). This will be the basis of our analysis, and it also allows drawing connections to other algorithms, originally not introduced as using a KL regularization. All the technical details are provided in the Appendix, but we give an intuition here, for the case $\tau = 0$ (no entropy). Let $\pi_{k+1} = \mathcal{G}_{\pi_k}^{\lambda,0}(q_k)$. This optimization problem can be solved analytically, yielding $\pi_{k+1} \propto \pi_k \exp \frac{q_k}{\lambda}$. By direct induction, $\pi_0$ being uniform, we have $\pi_{k+1} \propto \pi_k \exp \frac{q_k}{\lambda} \propto \cdots \propto \exp \frac{1}{\lambda} \sum_{j=0}^{k} q_j$. This means that penalizing the greedy step with a KL divergence provides a policy being a softmax over the scaled sum of all past $q$-functions (no matter how they are obtained). This is reminiscent of dual averaging in convex optimization, hence the name.

We now introduce the Dual Averaging MPI (DA-MPI) scheme. Contrary to MD-MPI, we have to distinguish the cases $\tau = 0$ and $\tau \neq 0$. DA-MPI($\lambda,0$) and DA-MPI($\lambda,\tau > 0$) are

$$\begin{cases} \pi_{k+1} = \mathcal{G}^{0,\frac{\lambda}{k+1}}(h_k) \\ q_{k+1} = (T_{\pi_{k+1}|\pi_k}^{\lambda,0})^m q_k + \epsilon_{k+1} \\ h_{k+1} = \frac{k+1}{k+2}h_k + \frac{1}{k+2}q_{k+1} \end{cases} \text{and} \begin{cases} \pi_{k+1} = \mathcal{G}^{0,\tau}(h_k) \\ q_{k+1} = (T_{\pi_{k+1}|\pi_k}^{\lambda,\tau})^m q_k + \epsilon_{k+1} \\ h_{k+1} = \beta h_k + (1 - \beta)q_{k+1} \text{ with } \beta = \frac{\lambda}{\lambda+\tau} \end{cases} , \tag{2}$$

with $h_0 = q_0$ for $\tau = 0$ and $h_{-1} = 0$ for $\tau > 0$. The following result is proven in Appx. C.1.

**Proposition 1.** *For any $\lambda > 0$, MD-MPI($\lambda,0$) and DA-MPI($\lambda,0$) are equivalent (but not in the limit $\lambda \to 0$). Moreover, for any $\tau > 0$, MD-MPI($\lambda,\tau$) and DA-MPI($\lambda,\tau$) are equivalent.*

Table 1: Algorithms encompassed by MD/DA-MPI (in italic if new compared to [20]).

|  | only entropy | only KL | both |
|---|---|---|---|
| reg. eval. | Soft Q-learning [17, 21], SAC [22], *Mellowmax* [5] | DPP [7], *SQL* [6] | *CVI* [25], *AL* [9, 11] |
| unreg. eval. | *softmax DQN* [41] | TRPO [37], MPO [2], *Politex* [1], *MoVI* [43] | *softened LSPI* [31], *MoDQN* [43] |

**Links to existing algorithms.**   Equivalent schemes (1) and (2) encompass (possibly variations of) many existing RL algorithms (see Tab. 1 and details below). Yet, we think important to highlight that many of them don't consider regularization in the evaluation step (they use $T_{\pi_{k+1}}$ instead of $T_{\pi_{k+1}|\pi_k}^{\lambda,\tau}$), something we abbreviate as "*w/o*". If it does not

preclude convergence in the case $\tau = 0$ [20, Thm. 4], it is known for the case $\tau > 0$ and $\lambda = 0$ that the resulting Bellman operator may have multiple fixed points [5], which is not desirable. Therefore, we only consider a regularized evaluation for the analysis, but we will compare both approaches empirically. Now, we present the approaches encompassed by scheme (1) (see also Appx. B.1). Soft Actor Critic (SAC) [22] and soft Q-learning [21] are variations of MD-MPI($0,\tau$), as is softmax DQN [41] but *w/o.* The Mellowmax policy [5] is equivalent to MD-MPI($0,\tau$). TRPO and MPO are variations of MD-MPI($\lambda,0$), *w/o.* DPP [7] is almost a reparametrization of MD-MPI($\lambda,0$), and Conservative Value Iteration (CVI) [25] is a reparametrization of MD-MPI$_1$($\lambda,\tau$), which consequently also generalizes Advantage Learning (AL) [9, 11]. Next, we present the approaches encompassed by schemes (2) (see also Appx. B.2). Politex [1] is a PI scheme for the average reward case, building upon prediction with expert advice. In the discounted case, it is DA-MPI($\lambda,0$), *w/o.* Momentum Value Iteration (MoVI) [43] is a limit case of DA-MPI($\lambda,0$), *w/o.*, as $\lambda \to 0$, and its practical extension to deep RL momentum DQN (MoDQN) is a limit case of DA-MPI($\lambda,\tau$), *w/o.* SQL [6] is a limit case of DA-MPI($\lambda, 0$) as $\lambda \to 0$. Softened LSPI [30] deals with zero-sum Markov games, but specialized to single agent RL it is a limit case of DA-MPI($\lambda,\tau$), *w/o.*

## 4 Theoretical Analysis

Here, we analyze the propagation of errors of MD-MPI, through the equivalent DA-MPI, for the case $m = 1$ (that is regularized VI, the extension to $m > 1$ remaining an open question). We provide component-wise bounds that assess the quality of the learned policy, depending on $\tau = 0$ or not. From these, $\ell_p$-norm bounds could be derived, using [36, Lemma 5].

**Analysis of DA-VI($\lambda$,0).** This corresponds to scheme (2), left, with $m = 1$. The following Thm. is proved in Appx. C.2.

**Theorem 1.** *Define* $E_k = -\sum_{j=1}^k \epsilon_j$, $A_k^1 = (I - \gamma P_{\pi_*})^{-1} - (I - \gamma P_{\pi_k})^{-1}$ *and* $g^1(k) = \frac{4}{1-\gamma} \frac{v_{max}^\lambda}{k}$. *Assume that* $\|q_k\|_\infty \leq v_{max}$. *We have* $0 \leq q_* - q_{\pi_k} \leq \left| A_k^1 \frac{E_k}{k} \right| + g^1(k)\mathbf{1}$.

**Remark 1.** *The assumption* $\|q_k\|_\infty \leq v_{max}$ *is not strong. It can be enforced by simply clipping the result of the evaluation step in* $[-v_{max}, v_{max}]$. *See also Appx. C.3.*

To ease the discussion, we express an $\ell_\infty$-bound as a direct corollary of Thm. 1:

$$\|q_* - q_{\pi_k}\|_\infty \leq \frac{2}{1-\gamma} \left\| \frac{1}{k} \sum_{j=1}^k \epsilon_j \right\|_\infty + \frac{4}{1-\gamma} \frac{v_{\max}^\lambda}{k}.$$

We also recall the typical propagation of errors of AVI without regularization (*e.g.* [36], we scale the sum by $1 - \gamma$ to make explicit the normalizing factor of a discounted sum):

$$\|q_* - q_{\pi_k}\|_\infty \leq \frac{2\gamma}{(1-\gamma)^2} \left( (1-\gamma) \sum_{j=1}^k \gamma^{k-j} \|\epsilon_j\|_\infty \right) + \frac{2}{1-\gamma} \gamma^k v_{\max}.$$

For each bound, the first term can be decomposed as a factor, the *horizon term* ($(1-\gamma)^{-1}$ is the average horizon of the MDP), scaling the *error term*, that expresses how the errors made at each iteration reflect in the final performance. The second term reflects the influence of the initialization over iterations, without errors it give the *rate of convergence* of the algorithms. We discuss these three terms.

**Rate of convergence.** It is slower for DA-VI($\lambda$,0) than for AVI, $\gamma^k = o(\frac{1}{k})$. This was to be expected, as the KL term slows down the policy updates. It is not where the benefits of KL regularization arise. However, notice that for $k$ small enough and $\gamma$ close to 1, we may have $\frac{1}{k} \leq \gamma^k$. This term has also a linear dependency to $\lambda$ (through $v_{\max}^\lambda$), suggesting that a lower $\lambda$ is better. This is intuitive, a larger $\lambda$ leads to smaller changes of the policy, and thus to a slower convergence.

**Horizon term.** We have a linear dependency to the horizon, instead of a quadratic one, which is very strong. Indeed, it is known that the square dependency to the horizon is tight

for API and AVI [35]. The only algorithms based on ADP having a linear dependency we are aware of make use of non-stationary policies [35, 8], and have never led to practical (deep) RL algorithms. Minimizing directly the Bellman residual would also lead to a linear dependency (*e.g.*, [32, Thm. 1]), but it comes with its own drawbacks [19] (*e.g.*, bias problem with stochastic dynamics, and it is not used in deep RL, as far as we know).

**Error term.** For AVI, the error term is a discounted *sum of the norms* of the successive estimation errors, while in our case it is the *norm of the average* of these estimation errors. The difference is fundamental, it means that the KL regularization allows for a compensation of the errors made at each iteration. Assume that the sequence of errors is a martingale difference. AVI would not converge in this case, while DA-VI($\lambda$, 0) converges to the optimal policy ($\|\frac{1}{k}\sum_{j=1}^{k}\epsilon_j\|_\infty$ converges to 0 by the law of large numbers). As far as we know, only SQL and DPP have such an error term, but they have a worse dependency to the horizon.

Thm. 1 is the first result showing that an RL algorithm can benefit from both a linear dependency to the horizon and from an averaging of the errors, and we argue that this explains, at least partially, the beneficial effect of using a KL regularization. Notice that Thm. 4 of Geist et al. [20] applies to DA-VI($\lambda$, 0), as they study more generally MPI regularized by a Bregman divergence. Although they bound a regret rather than $q_* - q_{\pi_k}$, their result is comparable to AVI, with a quadratic dependency to the horizon and a discounted sum of the norms of the errors. Therefore, our result significantly improves previous analyses.

We illustrate the bound with a simple experiment[2], see Fig. 1, left. We observe that AVI doesn't converge, while DA-VI($\lambda$,0) does, and that higher values of $\lambda$ slow down the convergence. Yet, they are also a bit more stable. This is not explained by our bound but is quite intuitive (policies changing less between iterations).

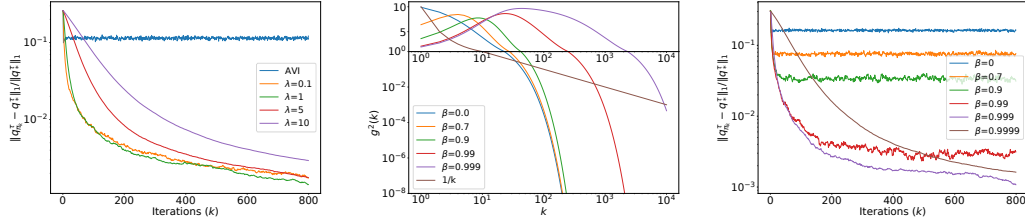

Figure 1: **Left**: behavior for Thm 1. **Middle**: function $g^2(k)$. **Right**: behavior for Thm 2.

**Analysis of DA-VI($\lambda$,$\tau$).** This is scheme (2), right, with $m = 1$. Due to the non-vanishing entropy term in the greedy step, it cannot converge to the unregularized optimal $q$-function. Yet, without errors and with $\lambda = 0$, it would converge to the solution of the MDP regularized by the scaled entropy (that is, considering the reward augmented by the scaled entropy). Our bound will show that adding a KL penalty does not change this. To do so, we introduce a few notations. The proofs of the following claims can be found in [20], for example. We already have defined the operator $T_\pi^{0,\tau}$. It has a unique fixed point, that we write $q_\pi^\tau$. The unique optimal $q$-function is $q_*^\tau = \max_\pi q_\pi^\tau$. We write $\pi_*^\tau = \mathcal{G}^{0,\tau}(q_*^\tau)$ the associated unique optimal policy, and $q_{\pi_*^\tau}^\tau = q_*^\tau$. The next result is proven in Appx. C.4.

**Theorem 2.** *For a sequence of policies $\pi_0, \ldots, \pi_k$, we define $P_{k:j} = P_{\pi_k}P_{\pi_{k-1}}\ldots P_{\pi_j}$ if $j \le k$, $P_{k:j} = I$ else. We define $A_{k:j}^2 = P_{\pi_*^\tau}^{k-j} + (I - \gamma P_{\pi_{k+1}})^{-1}P_{k:j+1}(I - \gamma P_{\pi_j})$. We define $g^2(k) = \gamma^k(1 + \frac{1-\beta}{1-\gamma})\sum_{j=0}^{k}(\frac{\beta}{\gamma})^j v_{max}^\tau$, with $\beta$ as defined in Eq. (2). Finally, we define $E_k^\beta = (1-\beta)\sum_{j=1}^{k}\beta^{k-j}\epsilon_j$. With these notations: $0 \le q_*^\tau - q_{\pi_{k+1}}^\tau \le \sum_{j=1}^{k}\gamma^{k-j}\left|A_{k:j}^2 E_j^\beta\right| + g^2(k)\mathbf{1}$.*

Again, to ease the discussion, we express an $\ell_\infty$-bound as a direct corollary of Thm. 2:

$$\|q_*^\tau - q_{\pi_{k+1}}^\tau\|_\infty \leq \frac{2}{(1-\gamma)^2} \left( (1-\gamma) \sum_{j=1}^{k} \gamma^{k-j} \|E_j^\beta\|_\infty \right) + \gamma^k (1 + \frac{1-\beta}{1-\gamma}) \sum_{j=0}^{k} \left(\frac{\beta}{\gamma}\right)^j v_{\max}^\tau.$$

There is a square dependency to the horizon, as for AVI. We discuss the other terms.

**Rate of convergence.** It is given by the function $g^2$, defined in Thm. 2. If $\beta = \gamma$, we have $g^2(k) = 2(k+1)\gamma^k v_{\max}^\tau$. If $\beta \neq \gamma$, we have $g^2(k) = (1 + \frac{1-\beta}{1-\gamma})\frac{\beta^{k+1} - \gamma^{k+1}}{\beta - \gamma}$. In all cases, $g^2(k) = o(\frac{1}{k})$, so it is asymptotically faster than in Thm. 1, but the larger the $\beta$, the slower the initial convergence. This is illustrated in Fig. 1, middle (notice that it's a logarithmic plot, except for the upper part of the $y$-axis).

**Error rate.** As with AVI, the error term is a discounted sum of the norms of errors. However, contrary to AVI, each error term is not an iteration error, but a moving average of past iteration errors, $E_k^\beta = \beta E_{k-1}^\beta + (1-\beta)\epsilon_k$. In the ideal case where the sequence of these errors is a martingale difference with respect to the natural filtration, this term no longer vanishes, contrary to $\frac{1}{k}E_k$. However, it can reduce the variance. For simplicity, assume that the $\epsilon_j$'s are i.i.d. of variance 1. In this case, it is easy to see that the variance of $E_k^\beta$ is bounded by $1 - \beta < 1$, that tends toward 0 for $\beta$ close to 1. Therefore, we advocate that DA-VI$_1(\lambda, \tau)$ allows for a better control of the error term than AVI (retrieved for $\beta = 0$). Notice that if asymptotically this error term predominates, the non-asymptotic behavior is also driven by the convergence rate $g^2$, which will be faster for $\beta$ closer to 0. Therefore, there is a trade-off, illustrated in Fig. 1, right (for the same simple experiment[2]). Higher values of $\beta$ lead to better asymptotic performance, but at the cost of slower initial convergence rate.

**Interplay between the KL and the entropy terms.** The l.h.s. of the bound of Thm. 2 solely depends on the entropy scale $\tau$, while the r.h.s. solely depends on the term $\beta = \frac{\lambda}{\lambda + \tau}$. DA-VI$(\lambda, \tau)$ approximates the optimal policy of the regularized MDP, while we are usually interested in the solution of the original one. We have that $\|q_* - q_{\pi_*^\tau}\|_\infty \leq \frac{\tau \ln|\mathcal{A}|}{1-\gamma}$ [20], this bias can be controlled by setting an (arbitrarily) small $\tau$. This does not affect the r.h.s. of the bound, as long as the scale of the KL term follows (such that $\frac{\lambda}{\lambda + \tau}$ remains fixed to the chosen value). So, Thm. 2 suggests to set $\tau$ to a very small value and to choose $\lambda$ such that we have a given value of $\beta$. However, adding an entropy term has been proven efficient empirically, be it with arguments of exploration and robustness [22] or regarding the optimization landscape [3]. Our analysis does not cover this aspect. Indeed, it applies to $\lambda = \beta = 0$ (that is, solely entropy regularization), giving the propagation of errors of SAC, as a special case of [20, Thm. 3]. In this case, we retrieve the bound of AVI ($E_j^0 = \epsilon_j$, $g^2(k) \propto \gamma^k$), up to the bounded quantity. Thus, it does not show an advantage of using solely an entropy regularization, but it shows the advantage for considering an additional KL regularization, if the entropy is of interest for other reasons.

We end this discussion with some related works. The bound of Thm. 2 is similar to the one of CVI, despite a quite different proof technique. Notably, both involve a moving average of the errors. This is not surprising, CVI being a reparameterization of DA-VI. The core difference is that by bounding the distance to the regularized optimal $q$-function (instead of the unregularized one), we indeed show to what the algorithm converges without error. Shani et al. [40] study a variation of TRPO, for which they show a convergence rate of $\mathcal{O}(\frac{1}{\sqrt{k}})$, improved to $\mathcal{O}(\frac{1}{k})$ when an additional entropy regularizer is considered. This is to be compared to the convergence rate of our variation of TRPO, $\mathcal{O}(\frac{1}{k}) = o(\frac{1}{\sqrt{k}})$ (Thm. 1) improved to $g^2(k) = o(\frac{1}{k})$ with an additional entropy term (Thm. 2). Our rates are much better. However, this is only part of the story. We additionally show a compensation of errors in both cases, something not covered by their analysis. They also provide a sample complexity, but it is much worse than the one of SQL, that we would improve (thanks to the improved horizon term). Therefore, our results are stronger and more complete.

**Limitations of our analysis.** Our analysis provides strong theoretical arguments in favor of considering KL regularization in RL. Yet, it has also some limitations. First, it does not

provide arguments for using only entropy regularization, as already extensively discussed (even though it provides arguments for combining it with a KL regularization). Second, we study how the errors propagate over iterations, and show that KL allows for a compensation of these errors, but we say nothing about how to control these errors. This depends heavily on how the $q$-functions are approximated and on the data used to approximate them. We could easily adapt the analysis of Azar et al. [6] to provide sample complexity bounds for MD-VI in the case of a tabular representation and with access to a generative model, but providing a more general answer is difficult, and beyond the scope of this paper. Third, we assumed that the greedy step was performed exactly. This assumption would be reasonable with a linear parameterization and discrete actions, but not if the policy and the $q$-function are approximated with neural networks. In this case, the equivalence between MD-VI and DA-VI no longer holds, suggesting various ways of including the KL regularizer (explicitly, MD-VI, or implicitly, DA-VI). Therefore, we complement our thorough theoretical analysis with an extensive empirical study, to analyse the core effect of regularization in deep RL.

## 5   Empirical study

Before all, we would like to highlight that if regularization is a core component of successful deep RL algorithms (be it with entropy, KL, or both), it is never the sole component. For example, SAC uses a twin critic [18], TRPO uses a KL hard constraint rather than a KL penalty [39], or MPO uses retrace [29] for value function evaluation. All these further refinements play a role in the final performance. On the converse, our goal is to study the core effect of regularization, especially of KL regularization, in a deep RL context. To achieve this, we notice that DA-VI and MD-VI are extensions of AVI. One of the most prevalent VI-based deep RL algorithm being DQN [28], our approach is to start from a reasonably tuned version of it [15] and to provide the minimal modifications to obtain deep versions of MD-VI or DA-VI. Notably, we fixed the meta-parameters to the best values for DQN.

**Practical algorithms.**   We describe briefly the variations we consider, a complementary high-level view is provided in Appx. E.1 and all practical details in Appx. E.2. We modify DQN by adding an actor. For the **evaluation step**, we keep the DQN loss, modified to account for regularization (that we'll call "$w/$", and that simply consists in adding the regularization term to the target $q$-network). Given that many approaches ignore the regularization there, we'll also consider the DQN loss (denoted "$w/o$" before, not covered by our analysis). For the **greedy step**, MD-VI and DA-VI are no longer equivalent. For MD-VI, there are two ways of approximating the regularized policy. The first one, denoted "*MD direct*", consists in directly solving the optimization problem corresponding to the regularized greediness, the policy being a neural network. This is reminiscent of TRPO (with a penalty rather than a constraint). The second one, denoted "*MD indirect*", consists in computing the analytical solution to the greedy step ($\pi_{k+1} \propto \pi_k^\beta \exp(\frac{1}{\lambda}\beta q_k)$) and to approximate it with a neural network. This is reminiscent of MPO. For DA-VI, we have to distinguish $\tau > 0$ from $\tau = 0$. In the first case, the regularized greedy policy can be computed analytically from an $h$-network, that can be computed by fitting a moving average of the online $q$-network and of a target $h$-network. This is reminiscent of MoDQN. If $\tau = 0$, DA-VI($\lambda$,0) is not practical in a deep learning setting, as it requires averaging over iterations. Updates of target networks are too fast to consider them as new iterations, and a moving average is more convenient. So, we only consider the limit case $\lambda, \tau \to 0$ with $\beta = \frac{\lambda}{\lambda+\tau}$ kept constant. This is MoDQN with fixed $\beta$, and the evaluation step is necessarily unregularized ($\lambda = \tau = 0$). To sum up, we have six variations (three kinds of greediness, evaluation regularized or not), restricted to five variations for $\tau = 0$.

**Research questions.**   Before describing the empirical study, we state the research questions we would like to address. The first is to know if regularization, without further refinements, helps, compared to the baseline DQN. The second one is to know if adding regularization in the evaluation step, something required by our analysis, provides improved empirical results. The third one is to compare the different kinds of regularized greediness, which are no longer equivalent with approximation. The last one is to study the effect of entropy, not covered by our analysis, and its interplay with the KL term.

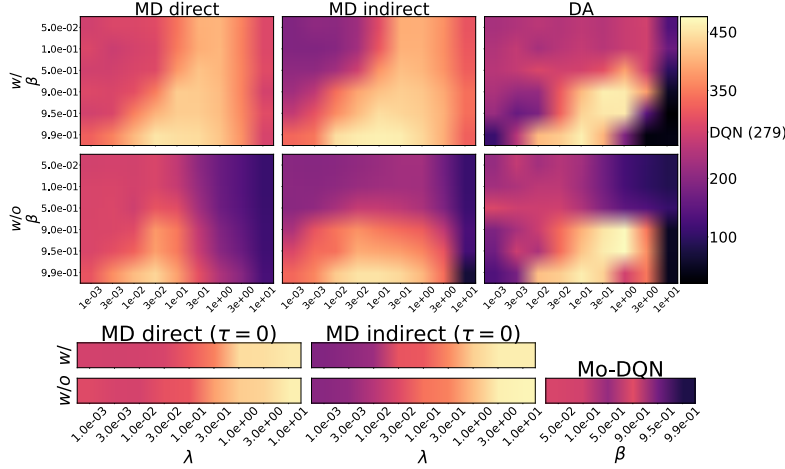

Figure 2: Cartpole.

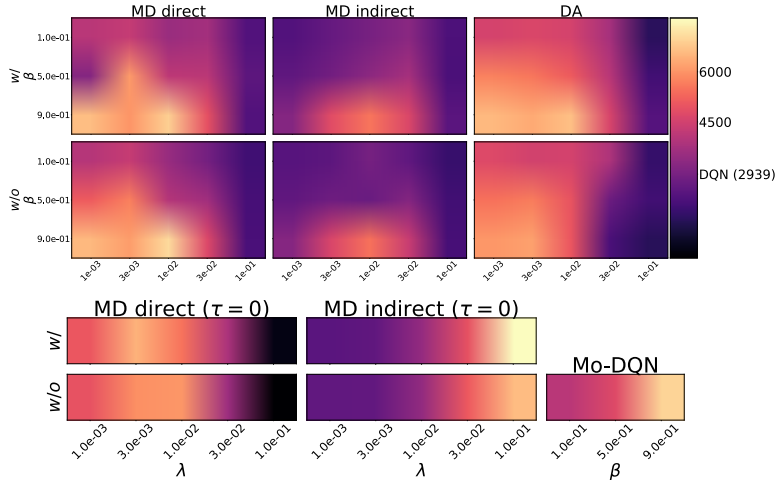

Figure 3: Asterix.

**Environments.** We consider two environments here (more are provided in Appx. E). The light Cartpole from Gym [14] allows for a large sweep over the parameters, and to average each result over 10 seeds. We also consider the Asterix Atari game [10], with sticky actions, to assess the effect of regularization on a large-scale problem. The sweep over parameters is smaller, and each result is averaged over 3 seeds.

**Visualisation.** For each environment, we present results as a table, the rows corresponding to the type of evaluation ($w/$ or $w/o$), the columns to the kind of greedy step. Each element of this table is a grid, varying $\beta$ for the rows and $\tau$ for the columns. One element of this grid is the average undiscounted return per episode obtained during training, averaged over the number of seeds. On the bottom of this table, we show the limit cases with the same principle, varying with $\lambda$ for MD-VI and with $\beta$ for DA-VI (ony $w/o$, as explained before). The scale of colors is common to all these subplots, and the performance of DQN is indicated on this scale for comparison. Additional visualisations are provided in Appx. E.

**Discussion.** Results are provided in Fig. 2 and 3. First, we observe that regularization helps. Indeed, the results obtained by all these variations are better than the one of DQN, the baseline, for a large range of the parameters, sometime to a large extent. We also observe that, for a given value of $\tau$, the results are usually better for medium to large values of

$\beta$ (or $\lambda$), suggesting that **KL regularization is beneficial** (even though too large KL regularization can be harmful in some case, for example for MD direct, $\tau = 0$, on Asterix).

Then, we study the effect of regularizing the evaluation step, something suggested by our analysis. The effect of this can be observed by comparing the first row to the second row of each table. One can observe that the range of good parameters is larger in the first row (especially for large entropy), suggesting that **regularizing the evaluation step helps**. Yet, we can also observe that when $\tau = 0$ (no entropy), there is much less difference between the two rows. This suggests that adding the entropy regularization to the evaluation step might be more helpful (but adding the KL term too is costless and never harmful).

Next, we study the effect of the type of greediness. MD-direct shows globally better results than MD-indirect, but MD-indirect provides the best result on both environments (by a small margin), despite being more sensitive to the parameters. DA is more sensitive to parameters than MD for Cartpole, but less for Asterix, its best results being comparable to those of MD. This let us think that **the best choice of greediness is problem dependent**, something that goes beyond our theoretical analysis.

Last, we discuss the effect of entropy. As already noticed, for a given level of entropy, medium to large values of the KL parameter improve performance, suggesting that entropy works better in conjunction with KL, something appearing in our bound. Now, observing the table corresponding to $\tau = 0$ (no entropy), we observe that we can obtain comparable best performance with solely a KL regularization, especially for MD. This suggests that **entropy is better with KL, and KL alone might be sufficient**. We already explained that some beneficial aspects of entropy, like exploration or better optimization landscape, are not explained by our analysis. However, we hypothesize that KL might have similar benefits. For examples, entropy enforces stochastic policies, which helps for exploration. KL has the same effect (if the initial policy is uniform), but in an adaptive manner (exploration decreases with training time).

## 6 Conclusion

We provided an explanation of the effect of KL regularization in RL, through the implicit averaging of $q$-values. We provided a very strong performance bound for KL regularization, the very first RL bound showing both a linear dependency to the horizon and an averaging the estimation errors. We also analyzed the effect of KL regularization with an additional entropy term. The introduced abstract framework encompasses a number of existing approaches, but some assumptions we made do not hold when neural networks are used. Therefore, we complemented our thorough theoretical analysis with an extensive empirical study. It confirms that KL regularization is helpful, and that regularizing the evaluation step is never detrimental. It also suggests that KL regularization alone, without entropy, might be sufficient (and better than entropy alone).

The core issue of our analysis is that it relies heavily on the absence of errors in the greedy step, something we deemed impossible with neural networks. However, Vieillard et al. [42] proposed subsequently a reparameterization of our regularized approximate dynamic scheme. The resulting approach, called "Munchausen Reinforcement Learning", is simple and general, and provides agents outperforming the state of the art. Crucially, thanks to this reparameterization, there's no error in their greedy step and our bounds apply readily. More details can be found in [42].

**Broader impact.** Our core contribution is theoretical. We unify a large body of the literature under KL-regularized reinforcement learning, and provide strong performance bounds, among them the first one ever to combine a linear dependency to the horizon and an averaging of the errors. We complement these results with an empirical study. It shows that the insights provided by the theory can still be used in a deep learning context, when some of the assumptions are not satisfied. As such, we think the broader impact of our contribution to be the same as the one of reinforcement learning.

**Funding transparency statement.** Nothing to disclose.

## Footnotes

[2] We illustrate the bounds in a simple tabular setting with access to a generative model. Considering random MDPs (called Garnets), at each iteration of DA-VI we sample a single transition for each state-action couple and apply the resulting sampled Bellman operator. The error $\epsilon_k$ is the difference between the sampled and the exact operators. The sequence of these estimation errors is thus a martingale difference w.r.t. its natural filtration [6] (one can think about bounded, centered and roughly independent errors). More details about this practical setting are provided in Appx. D.

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
