[Supplementary Material]

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

**Content.** These appendices complement the core paper with the following:

- Appx. A is a warm-up that states a few facts about the Legendre-Fenchel transform, useful all along the derivations.
- Appx. B justifies the connections drawn in Sec. 3 between MD-MPI or DA-MPI and the literature.
- Appx. C provides the proofs of all stated theoretical results, as well as some necessary lemmata.
- Appx. D provides details about the experiment used to illustrate the bounds in Sec. 4.
- Appx. E provides additional details regarding the practical algorithms and the experiments, as well as additional experiments and visualisations.

## A Convex Conjugacy for KL and Entropy Regularization

Let $q \in \mathbb{R}^{\mathcal{S} \times \mathcal{A}}$ and $\mu \in \Delta_{\mathcal{A}}^{\mathcal{S}}$, and consider the general greedy step $\pi' \in \mathcal{G}_{\mu}^{\lambda,\tau}$, the optimization being understood here state-wise.

$$\pi' \in \underset{\pi \in \Delta_{\mathcal{A}}^{\mathcal{S}}}{\operatorname{argmax}} \left( \langle \pi, q \rangle - \lambda \operatorname{KL}(\pi \| \mu) + \tau \mathcal{H}(\pi) \right). \tag{3}$$

The function $\lambda \operatorname{KL}(\pi \| \mu) - \tau \mathcal{H}(\pi)$ being convex in $\pi$, this optimization problem is related to the Legendre-Fenchel transform (*e.g.*, Hiriart-Urruty and Lemaréchal [23, Ch. E]), or convex conjugate (which is the maximum rather than the maximizer). First, we consider a simple case, $\lambda = 0$ and $\tau = 1$. It is well known in this case that the maximum (the convex conjugate) is the log-sum-exp function and the maximizer (the gradient of the convex conjugate) is the softmax (*e.g.*, Boyd and Vandenberghe [12, Ex. 3.25]):

$$\max_{\pi \in \Delta_{\mathcal{A}}^{\mathcal{S}}} \left( \langle \pi, q \rangle + \mathcal{H}(\pi) \right) = \ln \langle \mathbf{1}, \exp q \rangle \in \mathbb{R}^{\mathcal{S}},$$

$$\underset{\pi \in \Delta_{\mathcal{A}}^{\mathcal{S}}}{\operatorname{argmax}} \left( \langle \pi, q \rangle + \mathcal{H}(\pi) \right) = \frac{\exp q}{\langle \mathbf{1}, \exp q \rangle} \in \mathbb{R}^{\mathcal{S} \times \mathcal{A}},$$

with $\mathbf{1} \in \mathbb{R}^{\mathcal{S} \times \mathcal{A}}$ the vector of which all components are equal to 1. We made use of the notations introduced in Sec. 2, and overload $v \in \mathbb{R}^{\mathcal{S}}$ to $v \in \mathbb{R}^{\mathcal{S} \times \mathcal{A}}$ as $v(s,a) = v(s)$. To make things clear, it gives

$$[\ln \langle \mathbf{1}, \exp q \rangle](s) = \ln \sum_{a \in \mathcal{A}} \exp q(s,a)$$

$$\text{and} \quad \left[ \frac{\exp q}{\langle \mathbf{1}, \exp q \rangle} \right](s,a) = \frac{\exp q(s,a)}{\sum_{a' \in \mathcal{A}} q(s,a')}.$$

Notice also that a direct consequence of this is that

$$\ln \langle \mathbf{1}, \exp q \rangle = \langle \pi', q \rangle + \mathcal{H}(\pi') \text{ with } \pi' = \frac{\exp q}{\langle \mathbf{1}, \exp q \rangle}.$$

From this simple case, we can easily handle the general case. We have

$$\langle \pi, q \rangle - \lambda \operatorname{KL}(\pi \| \mu) + \tau \mathcal{H}(\pi) = \langle \pi, q \rangle - \lambda \langle \pi, \ln \pi - \ln \mu \rangle - \tau \langle \pi, \ln \pi \rangle$$

$$= \langle \pi, q + \lambda \ln \mu \rangle - (\lambda + \tau) \langle \pi, \ln \pi \rangle$$

$$= (\lambda + \tau) \left( \left\langle \pi, \frac{q + \lambda \ln \mu}{\lambda + \tau} \right\rangle + \mathcal{H}(\pi) \right).$$

From this, we can deduce directly that the maximum of (3) is

$$\max_{\pi \in \Delta_{\mathcal{A}}^{\mathcal{S}}} \left( \langle \pi, q \rangle - \lambda \operatorname{KL}(\pi \| \mu) + \tau \mathcal{H}(\pi) \right) = (\lambda + \tau) \ln \left\langle 1, \exp \frac{q + \lambda \ln \mu}{\lambda + \tau} \right\rangle$$

$$= (\lambda + \tau) \ln \left\langle \mu^{\frac{\lambda}{\lambda+\tau}}, \exp \frac{q}{\lambda + \tau} \right\rangle \tag{4}$$

$$= (\lambda + \tau) \left( \ln \sum_{a \in \mathcal{A}} \mu(a|s)^{\frac{\lambda}{\lambda+\tau}} \exp \frac{q(s,a)}{\lambda + \tau} \right)_{s \in \mathcal{S}},$$

and that the maximizer of (3) is

$$\operatorname*{argmax}_{\pi\in\Delta_{\mathcal{A}}^{\mathcal{S}}}\left(\langle\pi,q\rangle-\lambda\operatorname{KL}(\pi||\mu)+\tau\mathcal{H}(\pi)\right)=\frac{\exp\frac{q+\lambda\ln\mu}{\lambda+\tau}}{\langle\mathbf{1},\exp\frac{q+\lambda\ln\mu}{\lambda+\tau}\rangle}=\frac{\mu^{\frac{\lambda}{\lambda+\tau}}\exp\frac{q}{\lambda+\tau}}{\langle\mathbf{1},\mu^{\frac{\lambda}{\lambda+\tau}}\exp\frac{q}{\lambda+\tau}\rangle} \tag{5}$$

$$=\left(\frac{\mu(a|s)^{\frac{\lambda}{\lambda+\tau}}\exp\frac{q(s,a)}{\lambda+\tau}}{\sum_{a'\in\mathcal{A}}\mu(a'|s)^{\frac{\lambda}{\lambda+\tau}}\exp\frac{q(s,a')}{\lambda+\tau}}\right)_{(s,a)\in\mathcal{S}\times\mathcal{A}}$$

Again, the relationship between the maximum and the maximizer gives

$$(\lambda+\tau)\ln\left\langle\mu^{\frac{\lambda}{\lambda+\tau}},\exp\frac{q}{\lambda+\tau}\right\rangle=\langle\pi',q\rangle-\lambda\operatorname{KL}(\pi'||\mu)+\tau\mathcal{H}(\pi') \tag{6}$$

$$\text{with }\pi'=\frac{\mu^{\frac{\lambda}{\lambda+\tau}}\exp\frac{q}{\lambda+\tau}}{\langle\mathbf{1},\mu^{\frac{\lambda}{\lambda+\tau}}\exp\frac{q}{\lambda+\tau}\rangle}.$$

# B   Connections to existing algorithms

In this section, we justify the connections stated in Sec. 3 between the considered regularized ADP schemes and the literature.

## B.1   Connection of MD-MPI($\lambda,\tau$) to other algorithms

**Connection to SAC.**   We stated that SAC [22] is a variation of MD-MPI($0,\tau$). SAC was introduced as PI scheme ($m=\infty$), while it is practically implemented as VI scheme ($m=1$). We keep the VI viewpoint for this discussion. The MD-VI($0,\tau$) scheme is given by

$$\begin{cases} \pi_{k+1}=\mathcal{G}^{0,\tau}(q_k) \\ q_{k+1}=T_{\pi_{k+1}}^{0,\tau}q_k+\epsilon_{k+1} \end{cases}. \tag{7}$$

The regularized Bellman operator can be rewritten as follows:

$$T_{\pi_{k+1}}^{0,\tau}q_k=T_{\pi_{k+1}}q_k+\gamma P\tau\mathcal{H}(\pi_{k+1})=r+\gamma P\left(\langle\pi_{k+1},q_k\rangle-\tau\langle\pi_{k+1},\ln\pi_{k+1}\rangle\right)$$

$$=r+\gamma P\langle\pi_{k+1},q_k-\tau\ln\pi_{k+1}\rangle.$$

This is exactly the Bellman operator considered in SAC. For the greedy step, we have directly from Eq. (5) that $\pi_{k+1}\propto\exp\frac{q_k}{\tau}$. In SAC, continuous actions are considered, so the policy cannot be computed (due to the partition function). Therefore, it is approximated with a neural network by minimizing a reverse KL divergence (that allows getting rid of the partition function) between the neural policy and the target policy (the solution of the original greedy step):

$$\pi_{k+1}=\operatorname*{argmin}_{\pi_\theta}\mathbb{E}_s[\operatorname{KL}(\pi_\theta||\pi_{k+1}^*)]=\operatorname*{argmin}_{\pi_\theta}\mathbb{E}_s[\operatorname{KL}(\pi_\theta||\exp\frac{q}{\tau})]\text{ with }\pi_{k+1}^*=\frac{\exp\frac{q_k}{\tau}}{\langle\mathbf{1},\exp\frac{q_k}{\tau}\rangle}.$$

**Connection to Soft Q-learning.**   We stated that Soft Q-learning [17, 21] is also a variation of MD-MPI($0,\tau$). It is indeed a VI scheme, so a variation of MD-VI($0,\tau$) depicted in Eq. (7). As a direct consequence of Eq. (6), $\pi_{k+1}\propto\exp\frac{q_k}{\tau}$ being the maximizer, we have

$$\langle\pi_{k+1},q_k\rangle+\tau\mathcal{H}(\pi_{k+1})=\tau\ln\langle\mathbf{1},\exp\frac{q_k}{\tau}\rangle.$$

This allows rewriting the evaluation step as follows:

$$q_{k+1}=T_{\pi_{k+1}}^{0,\tau}q_k+\epsilon_{k+1}$$

$$=r+\gamma P\left(\langle\pi_{k+1},q_k\rangle+\tau\mathcal{H}(\pi_{k+1})\right)+\epsilon_{k+1}$$

$$\Leftrightarrow q_{k+1}=r+\gamma P\left(\tau\ln\langle\mathbf{1},\exp\frac{q_k}{\tau}\rangle\right)+\epsilon_{k+1}. \tag{8}$$

Eq. (8) is equivalent to Eq. (7), and it is the Bellman operator upon which Soft Q-learning is built (replacing the hard maximum by the log-sum-exp). Haarnoja et al. [21] additionally handle continuous actions, which requires some refinements.

**Connection to Softmax DQN.** We stated that Softmax DQN [41] is a variation of MD-MPI(0,$\tau$), but *w/o* (without regularization in the evaluation step). Therefore, it is scheme (7), but replacing $T^{0,\tau}_{\pi_{k+1}}$ by $T_{\pi_{k+1}}$:

$$\begin{cases} \pi_{k+1} = \mathcal{G}^{0,\tau}(q_k) \\ q_{k+1} = T_{\pi_{k+1}} q_k + \epsilon_{k+1} \end{cases}.$$

Given that $\pi_{k+1} \propto \exp \frac{q_k}{\tau}$, this amounts to iterating the following so called softmax operator

$$q_{k+1} = T_{\pi_{k+1}} q_k + \epsilon_{k+1}$$
$$= r + \gamma P \left\langle \frac{\exp \frac{q_k}{\tau}}{\langle \mathbf{1}, \exp \frac{q_k}{\tau} \rangle}, q_k \right\rangle + \epsilon_{k+1},$$

which is the core update rule of softmax DQN. Notice that this operator might not be a contraction (depending on the value of $\tau$), and that it can have multiple fixed points [5].

**Connection to the mellowmax policy.** Asadi and Littman [5] introduced a so-called mellowmax policy as a convergent alternative to the softmax operator. This can be indeed seen as an alternative way of regularizing the evaluation step. We explain here why. To do so, we reframe the mellowmax idea with our notations. Asadi and Littman [5] introduced the mellowmax operator as

$$\mathrm{mm}_\tau(q) = \tau \ln \left\langle \mathbf{1}, \frac{1}{|\mathcal{A}|} \exp \frac{q}{\tau} \right\rangle.$$

One can easily see that it is indeed the convex conjugate of the KL with respect to the uniform policy (that behaves like the entropy). Indeed, from Eq. (4), we have directly that

$$\mathrm{mm}_\tau(q) = \max_{\pi \in \Delta^{\mathcal{S}}_{\mathcal{A}}} \left( \langle \pi, q \rangle - \tau \, \mathrm{KL}(\pi || \pi_U) \right),$$

with $\pi_U$ the uniform policy. From Geist et al. [20], we know that the following equivalent schemes,

$$\begin{cases} \pi_{k+1} = \mathrm{argmax}_{\pi \in \Delta^{\mathcal{S}}_{\mathcal{A}}} \left( \langle \pi, q \rangle - \tau \, \mathrm{KL}(\pi || \pi_U) \right) \\ q_{k+1} = T_{\pi_{k+1}} q_k - \gamma P \tau \, \mathrm{KL}(\pi_{k+1} || \pi_U) \end{cases} \Leftrightarrow q_{k+1} = r + \gamma P \, \mathrm{mm}_\tau(q_k),$$

are convergent (MDP regularized with $\tau \, \mathrm{KL}(\cdot || \pi_U)$, the equivalence being from Eq. (6)). This is not the viewpoint of Asadi and Littman [5]. They try to find a policy $\pi'_{k+1}$ such that $q_{k+1} = r + \gamma P \, \mathrm{mm}_\tau(q_k) = r + \gamma P \langle \pi'_{k+1}, q_k \rangle$. To account for the possible existence of multiple policies, they look for the one with maximal entropy and solve (numerically) for

$$\pi'_{k+1} = \max_{\pi \in \Delta^{\mathcal{S}}_{\mathcal{A}} : \langle \pi, q_k \rangle = \mathrm{mm}_\tau(q_k)} \mathcal{H}(\pi).$$

Then, they apply $q_{k+1} = r + \gamma P \langle \pi'_{k+1}, q_k \rangle$. If there is no error when computing $\pi'_{k+1}$, this is equivalent to adding the regularization to the evaluation step.

**Connection to TRPO.** We stated that TRPO [37] is a variation of MD-MPI($\lambda$, 0), *w/o*. More precisely, it is a variation of MD-PI($\lambda$, 0):

$$\begin{cases} \pi_{k+1} = \mathcal{G}^{\lambda,0}(q_k) \\ q_{k+1} = T^{\infty}_{\pi_{k+1}} q_k + \epsilon_k = q_{\pi_{k+1}} + \epsilon_k \end{cases}. \tag{9}$$

In TRPO, the *q*-function is evaluated using Monte Carlo rollouts. The greedy policy is approximated with a neural network by directly solving the expected greedy step:

$$\pi_{k+1} = \mathrm{argmin}_{\pi_\theta} \mathbb{E}_s [\langle \pi_\theta, q_k \rangle - \lambda \, \mathrm{KL}(\pi_\theta || \pi_k)].$$

TRPO is indeed a bit different, as it uses importance sampling to sample actions according to $\pi_k$ (which is especially useful for continuous actions, but does not change the objective function), it uses a constraint based on the KL rather than a regularization, and it considers the KL in the other direction:

$$\pi_{k+1} = \mathrm{argmin}_{\pi_\theta : \mathbb{E}_s [\mathrm{KL}(\pi_k || \pi_\theta)] \leq \epsilon} \mathbb{E}_s [\mathbb{E}_{a \sim \pi_k(.|s)} [\langle \frac{\pi_\theta}{\pi_k}, q_k \rangle]].$$

However, from an abstract viewpoint, TRPO is close to scheme (9).

**Connection to MPO.** We stated that MPO [2] is also a variation of MD-MPI($\lambda$, 0), *w/o*:

$$\begin{cases} \pi_{k+1} = \mathcal{G}^{\lambda,0}(q_k) \\ q_{k+1} = T^m_{\pi_{k+1}} q_k + \epsilon_k \end{cases}. \tag{10}$$

The evaluation step is done by combining a TD approach with eligibility traces (a geometric average of $m$-step returns), rather than using $m$-step returns (that amounts to using the $T^m_\pi$ operator). For the greedy step, the analytic solution can be computed for any state-action couple, and generalized to the whole state-action space by minimizing a KL between this analytical solution and a neural network:

$$\pi_{k+1} = \underset{\pi_\theta}{\operatorname{argmin}} \, \mathbb{E}_s[\text{KL}(\pi^*_{k+1}||\pi_\theta)] = \underset{\pi_\theta}{\operatorname{argmax}} \, \mathbb{E}_s[\mathbb{E}_{a \sim \pi^*_{k+1}(.|s)}[\ln \pi_\theta(a|s)]]$$

$$\text{with } \pi^*_{k+1} = \frac{\pi_k \exp \frac{q_k}{\lambda}}{\langle \mathbf{1}, \pi_k \exp \frac{q_k}{\lambda} \rangle}.$$

The greedy step of MPO is indeed a bit different, the algorithm being derived from an expectation-maximization principle based on a probabilistic inference view of RL. The term $\lambda$ is not fixed but learnt by the minimization of a convex dual function (coming from viewing the KL term as a constraint rather than a regularization), and an additional KL penalty is added (not necessarily redundant with the initial one, as the KL there is in the other direction):

$$\pi_{k+1} = \underset{\pi_\theta : \mathbb{E}_s[\text{KL}(\pi_k||\pi_\theta)] \le \epsilon}{\operatorname{argmax}} \mathbb{E}_s[\mathbb{E}_{a \sim \pi^*_{k+1}(.|s)}[\ln \pi_\theta(a|s)]].$$

However, from an abstract viewpoint, MPO is close to scheme (10).

**Connection to DPP.** We stated that DPP [7] is a variation of MD-MPI($\lambda$, 0). More precisely, it is close to be a reparameterization of MD-VI($\lambda$, 0), the difference being mainly the error term:

$$\begin{cases} \pi_{k+1} = \mathcal{G}^{\lambda,0}(q_k) \\ q_{k+1} = T^{\lambda,0}_{\pi_{k+1}} q_k + \epsilon_k \end{cases}. \tag{11}$$

To derive the DPP update rule from Eq. (11), we consider $\epsilon_k = 0$. The greedy policy is, according to (5),

$$\pi_{k+1} = \frac{\pi_k \exp \frac{q_k}{\lambda}}{\langle \mathbf{1}, \pi_k \exp \frac{q_k}{\lambda} \rangle}.$$

Define $v_{k+1}$ as (the second equality coming from Eq. (6))

$$v_{k+1} = \langle \pi_{k+1}, q_k \rangle - \lambda \text{KL}(\pi_{k+1}||\pi_k) = \lambda \ln\langle \pi_k, \exp \frac{q_k}{\lambda} \rangle.$$

With this, we have

$$q_{k+1} = T^{\lambda,0}_{\pi_{k+1}} q_k = r + \gamma P(\langle \pi_{k+1}, q_k \rangle - \lambda \text{KL}(\pi_{k+1}||\pi_k)) = r + \gamma P v_{k+1}$$

Let us define $\psi_{k+1} \in \mathbb{R}^{\mathcal{S} \times \mathcal{A}}$ as

$$\psi_{k+1} = \lambda \ln\left(\pi_k \exp \frac{q_k}{\lambda}\right) = r + \gamma P v_k + \lambda \ln \pi_k. \tag{12}$$

Thus, we have

$$\pi_k = \frac{\exp \frac{\psi_k}{\lambda}}{\langle \mathbf{1}, \exp \frac{\psi_k}{\lambda} \rangle} \tag{13}$$

$$\text{and } v_k = \lambda \ln\langle \mathbf{1}, \frac{\psi_k}{\lambda} \rangle. \tag{14}$$

Injecting Eqs. (13) and (14) into (12), we get

$$\psi_{k+1} = r + \gamma P \lambda \ln\langle \mathbf{1}, \frac{\psi_k}{\lambda} \rangle + \psi_k - \lambda \ln\langle \mathbf{1}, \frac{\psi_k}{\lambda} \rangle.$$

This is how DPP is justified from a DP viewpoint [7, Appx. A]. It is a bit different from the DPP algorithm analyzed by Azar et al. [7], for which $\ln\langle\mathbf{1}, \frac{\psi_k}{\lambda}\rangle$ is replaced by $\langle\pi_k, \psi_k\rangle$ (both terms being equal in the limit $\lambda \to 0$), and that consider an estimation error $\epsilon'_{k+1}$:

$$\psi_{k+1} = r + \gamma P\langle\pi_k, \psi_k\rangle + \psi_k - \langle\pi_k, \psi_k\rangle + \epsilon'_{k+1}.$$

We advocate that the error $\epsilon'_k$ is usually harder to control than $\epsilon_k$ (or equivalently that $q_k$ is easier to estimate than $\psi_k$), because the function $\psi_*$ (the optimal $\psi$-function for the MDP) is equal to $-\infty$ for any suboptimal action [7, Cor. 4].

**Connection to CVI.** We stated that CVI is a reparametrization of MD-VI($\lambda,\tau$), that we recall (without the error term, to do the reparameterization):

$$\begin{cases} \pi_{k+1} = \mathcal{G}^{\lambda,\tau}(q_k) \\ q_{k+1} = T^{\lambda,\tau}_{\pi_{k+1}}q_k \end{cases}.$$

We now show how to derive the CVI update rule from this. The regularized greedy policy is, thanks to Eq. (5), and writing $\beta = \frac{\lambda}{\lambda+\tau}$:

$$\pi_{k+1} = \frac{\pi_k^\beta \exp\frac{\beta q_k}{\lambda}}{\langle\mathbf{1}, \pi_k^\beta \exp\frac{\beta q_k}{\lambda}\rangle}.$$

Similarly to DPP, we can define $v_{k+1}$ as (still using Eq. (6) for the second equality):

$$v_{k+1} = \langle\pi_{k+1}, q_k\rangle - \lambda \operatorname{KL}(\pi_{k+1}||\pi_k) + \tau\mathcal{H}(\pi_{k+1}) = \frac{\lambda}{\beta}\ln\langle\pi_k^\beta, \exp\frac{\beta q_k}{\lambda}\rangle.$$

With this, we have

$$q_{k+1} = T^{\lambda,0}_{\pi_{k+1}}q_k = r + \gamma P(\langle\pi_{k+1}, q_k\rangle - \lambda \operatorname{KL}(\pi_{k+1}||\pi_k) + \tau\mathcal{H}(\pi_{k+1})) = r + \gamma Pv_{k+1}.$$

Let us define $\psi_{k+1} \in \mathbb{R}^{\mathcal{S}\times\mathcal{A}}$ as

$$\psi_{k+1} = \frac{\lambda}{\beta}\ln\left(\pi_k^\beta \exp\frac{\beta q_k}{\lambda}\right) = r + \gamma Pv_k + \lambda \ln\pi_k. \tag{15}$$

Thus, we have

$$\pi_k = \frac{\exp\frac{\beta\psi_k}{\lambda}}{\langle\mathbf{1}, \exp\frac{\beta\psi_k}{\lambda}\rangle} \tag{16}$$

$$\text{and } v_k = \frac{\lambda}{\beta}\ln\langle\mathbf{1}, \frac{\beta\psi_k}{\lambda}\rangle. \tag{17}$$

Injecting Eqs. (16) and (17) into (15), we get

$$\psi_{k+1} = r + \gamma P\frac{\lambda}{\beta}\ln\langle\mathbf{1}, \frac{\beta\psi_k}{\lambda}\rangle + \beta(\psi_k - \frac{\lambda}{\beta}\ln\langle\mathbf{1}, \frac{\beta\psi_k}{\lambda}\rangle).$$

This is exactly the CVI update rule. Notice that setting $\beta = 1$, i.e., $\tau = 0$ (no entropy term), we retrieve DPP (which was to be expected). As we obtain CVI, by considering $\lambda + \tau \to 0$ while keeping $\beta = \frac{\lambda}{\lambda+\tau}$ constant, we retrieve advantage learning in the limit [9, 11], that DA-VI($\lambda,\tau$) thus generalizes.

### B.2 Connection of DA-MPI($\lambda,\tau$) to other algorithms

**Connection to Politex.** Politex [1] addresses the average reward criterion. It is a PI scheme, up to the fact that the policy, instead of being greedy according to the last $q$-function, is softmax according to the sum of all past $q$-function. In the discounted reward case considered here, this is exactly DA-PI($\lambda,0$), *w/o* (without regularization in the evaluation step):

$$\begin{cases} \pi_{k+1} = \mathcal{G}^{0, \frac{\lambda}{k+1}}(h_k) \\ q_{k+1} = T^\infty_{\pi_{k+1}}q_k + \epsilon_{k+1} = q_{\pi_{k+1}} + \epsilon_{k+1} \\ h_{k+1} = \frac{k+1}{k+2}h_k + \frac{1}{k+2}q_{k+1} \end{cases}$$

Indeed, by definition $h_k = \frac{1}{k+1}\sum_{j=0}^{k} q_j$ and the greedy policy is

$$\pi_{k+1} = \mathcal{G}^{0,\frac{\lambda}{k+1}}(h_k) = \frac{\exp\frac{(k+1)h_k}{\lambda}}{\langle \mathbf{1}, \exp\frac{(k+1)h_k}{\lambda}\rangle} = \frac{\exp\frac{\sum_{j=0}^{k} q_j}{\lambda}}{\langle \mathbf{1}, \exp\frac{\sum_{j=0}^{k} q_j}{\lambda}\rangle}.$$

This is exactly the Politex algorithm, but for the discounted reward case (that changes how the $q$-function is defined, and thus estimated).

**Connection to MoVI.** MoVI [43] is a VI scheme, up to the fact that the policy, instead of being greedy according to the last $q$-function, is greedy according to the average of past $q$-functions. It is indeed is a limiting case of DA-VI($\lambda$, 0), $w/o$:

$$\begin{cases} \pi_{k+1} = \mathcal{G}^{0,\frac{\lambda}{k+1}}(h_k) \\ q_{k+1} = T_{\pi_{k+1}}q_k + \epsilon_{k+1} \\ h_{k+1} = \frac{k+1}{k+2}h_k + \frac{1}{k+2}q_{k+1} \end{cases}.$$

It is well known that the limit of a softmax, when the temperatures goes to zero, is the greedy policy: $\mathcal{G}^{0,\frac{\lambda}{k+1}}(h_k) \to \mathcal{G}(h_k)$ as $\lambda \to 0$. So, DA-VI($\lambda \to 0$, 0), $w/o$, is the following scheme,

$$\begin{cases} \pi_{k+1} \in \mathcal{G}(h_k) \\ q_{k+1} = T_{\pi_{k+1}}q_k + \epsilon_{k+1} \\ h_{k+1} = \frac{k+1}{k+2}h_k + \frac{1}{k+2}q_{k+1} \end{cases},$$

that is exactly MoVI. Notice that it is different from MD-VI($\lambda \to 0$, 0), $w/o$, which is AVI (see also Prop. 1).

**Connection to momentum DQN.** Momentum DQN [43] was introduced as a practical heuristic to MoVI, changing the exact average by a moving average (more amenable to optimization with deep networks). We show below that it is indeed a limiting case of DA-VI($\lambda,\tau$), $w/o$ (without regularized greedy step), that is:

$$\begin{cases} \pi_{k+1} = \mathcal{G}^{0,\tau}(h_k) \\ q_{k+1} = T_{\pi_{k+1}}q_k + \epsilon_{k+1} \\ h_{k+1} = \beta h_k + (1-\beta)q_{k+1} \text{ with } \beta = \frac{\lambda}{\lambda+\tau} \end{cases}.$$

Fix $\beta \in (0,1)$, we can consider $\lambda,\tau \to 0$ with $\beta = \frac{\lambda}{\lambda+\tau}$ kept constant. In this case, the regularized greedy operator tends to the usual greedy one: $\mathcal{G}^{0,\tau}(h_k) \to \mathcal{G}(h_k)$ as $\tau \to 0$. In the limit, we obtain the following scheme,

$$\begin{cases} \pi_{k+1} = \mathcal{G}(h_k) \\ q_{k+1} = T_{\pi_{k+1}}q_k + \epsilon_{k+1} \\ h_{k+1} = \beta h_k + (1-\beta)q_{k+1} \end{cases}$$

for a chosen $\beta$, which is exactly momentum DQN with fixed $\beta$.

**Connection to Speedy Q-learning.** We stated that Speedy Q-learning [6] is a limiting case of DA-VI($\lambda$,0), which we recall (without the error term here):

$$\begin{cases} \pi_{k+1} = \mathcal{G}^{0,\frac{\lambda}{k+1}}(h_k) \\ q_{k+1} = T_{\pi_{k+1}}^{\lambda,0}q_k \\ h_{k+1} = \frac{k+1}{k+2}h_k + \frac{1}{k+2}q_{k+1} \end{cases}.$$

As shown in Lemma 2 in Appx. C.2, we have

$$T_{\pi_{k+1}|\pi_k}^{\lambda,0}q_k = (k+1)T_{\pi_{k+1}}^{0,\frac{\lambda}{k+1}}h_k - kT_{\pi_k}^{0,\frac{\lambda}{k}}h_{k-1}.$$

With this, DA-VI$_1$($\lambda$,0) can be expressed solely in terms of $h_k$ and $\pi_k$:

$$\begin{cases} \pi_{k+1} = \mathcal{G}^{0,\frac{\lambda}{k+1}}(h_k) \\ h_{k+1} = \frac{k+1}{k+2}h_k + \frac{1}{k+2}\left((k+1)T_{\pi_{k+1}}^{0,\frac{\lambda}{k+1}}h_k - kT_{\pi_k}^{0,\frac{\lambda}{k}}h_{k-1}\right). \end{cases} \tag{18}$$

As before, as $\lambda \to 0$, the regularized greedy step tends to the greedy step, $\mathcal{G}^{0,\frac{\lambda}{k+1}}(h_k) \to \mathcal{G}(h_k)$. Regarding the evaluation step, we can write, by definition of the regularized Bellman operator and using Eq. (6),

$$T_{\pi_{k+1}}^{0,\frac{\lambda}{k+1}} h_k = r + \gamma P \left( \langle \pi_{k+1}, h_k \rangle + \frac{\lambda}{k+1} \mathcal{H}(\pi_{k+1}) \right)$$

$$= r + \gamma P \left( \frac{\lambda}{k+1} \ln \langle \mathbf{1}, \exp \frac{(k+1)h_k}{\lambda} \rangle \right).$$

It is a classical result that the convex conjugate of the entropy tends to the hard maximum as the associated temperature goes to zero. For any $s \in \mathcal{S}$,

$$\lim_{\lambda \to 0} \frac{\lambda}{k+1} \ln \sum_{a \in \mathcal{A}} \exp \frac{(k+1)h_k(s,a)}{\lambda} = \frac{1}{k+1} \max_{a \in \mathcal{A}} ((k+1)h_k(s,a)) = \max_{a \in \mathcal{A}} h_k(s,a).$$

Writing $T_*$ the Bellman optimality operator, defined as $T_* q = \max_\pi T_\pi q$, we thus have

$$\lim_{\lambda \to 0} T_{\pi_{k+1}}^{0,\frac{\lambda}{k+1}} h_k = T_* h_k.$$

Thus, writing the limit of scheme (18) as $\lambda \to 0$, we obtain

$$h_{k+1} = (1 - \frac{1}{k+2})h_k + \frac{1}{k+2} ((k+1)T_* h_k - kT_* h_{k-1}),$$

which is exactly the Speedy Q-learning update rule.

**Connection to softened LSPI.** [31] address the problem of learning a Nash equilibria in zero-sum Markov games. They show that state of the art algorithms can be derived by minimizing the norm of the (projected) Bellman residual using a Newton descent, and propose more stable algorithms by using instead a quasi-Newton descent. Single agent reinforcement learning is a special case of zero-sum Markov games, and in this case the algorithm they propose can be written as follows, in an abstract way[3]:

$$\begin{cases} \pi_{k+1} \in \mathcal{G}(h_k) \\ q_{k+1} = T_{\pi_{k+1}}^\infty q_k + \epsilon_{k+1} = q_{\pi_{k+1}} + \epsilon_{k+1} \\ h_{k+1} = \beta h_k + (1-\beta)q_{k+1} \end{cases}.$$

Using the same arguments as for the connection to momentum DQN, this is a limit case of DA-PI($\lambda,\tau$), $w/o$, as $\lambda, \tau \to 0$ with $\beta = \frac{\lambda}{\lambda+\tau}$ kept constant. It is also closely related to Politex (the policy is greedy instead of being softmax, moving average of the $q$-values instead of an average).

# C    Proofs of Theoretical Results

In this section, we prove the results stated in the paper.

## C.1    Proof of Proposition 1

**Sketch of proof.**    As explained in the paper, the optimization problem $\pi_{k+1} = \mathcal{G}_{\pi_k}^{\lambda,0}(q_k)$ can be solved analytically, yielding $\pi_{k+1} \propto \pi_k \exp \frac{q_k}{\lambda}$. By direct induction, $\pi_0$ being uniform, we have $\pi_{k+1} \propto \pi_k \exp \frac{q_k}{\lambda} \propto \cdots \propto \exp \frac{1}{\lambda} \sum_{j=0}^k q_j$. Thus, the policy is indeed softmax according to the sum of $q$-values. Defining $h_k$ as the average of past $q$-values basically provides the stated DA-VI($\lambda$,0). The case with an additionnal entropy term is a bit more involved, but the principle is the same.

**Proof.** We start by proving the equivalence for the case $\tau = 0$. Recall that we assumed, with little loss of generality, that $\pi_0$ is the uniform policy. We recall MD-MPI($\lambda$,0):

$$\begin{cases} \pi_{k+1} = \mathcal{G}_{\pi_k}^{\lambda,0}(q_k) \\ q_{k+1} = (T_{\pi_{k+1}|\pi_k}^{\lambda,0})^m q_k + \epsilon_{k+1} \end{cases} . \tag{19}$$

Let us define $h_0 = q_0$ and $h_k$ for $k \geq 1$ as the average of past $q$-functions.

$$h_k = \frac{1}{k+1}\sum_{j=0}^{k} q_j = \frac{k}{k+1} h_{k-1} + \frac{1}{k+1} q_k.$$

As a direct consequence of Eq. (5), we have that $\pi_{k+1} \propto \pi_k \exp\frac{q_k}{\lambda}$. By direct induction,

$$\pi_{k+1} \propto \pi_k \exp\frac{q_k}{\lambda} \propto \pi_{k-1}\exp\frac{q_k + q_{k-1}}{\lambda} \propto \cdots \propto \exp\frac{\sum_{j=0}^{k} q_j}{\lambda} = \exp\frac{(k+1)h_k}{\lambda}.$$

Still thanks to Eq. (5), this means that $\pi_{k+1}$ satisfies

$$\pi_{k+1} = \underset{\pi \in \Delta_{\mathcal{A}}^{\mathcal{S}}}{\operatorname{argmax}}\left(\langle \pi, h_k \rangle + \frac{\lambda}{k+1}\mathcal{H}(\pi)\right) = \mathcal{G}^{0,\frac{\lambda}{k+1}}(h_k).$$

This shows that Eq. (19) is equivalent to

$$\begin{cases} \pi_{k+1} = \mathcal{G}^{0,\frac{\lambda}{k+1}}(h_k) \\ q_{k+1} = (T_{\pi_{k+1}|\pi_k}^{\lambda,0})^m q_k + \epsilon_{k+1} \\ h_{k+1} = \frac{k+1}{k+2} h_k + \frac{1}{k+2} q_{k+1} \end{cases} , \tag{20}$$

which is DA-MPI($\lambda$,0), and this shows the first part of the result. In the limit $\lambda \to 0$, the regularized greediness becomes the usual greediness (hard maximum over $q$-values) and the (regularized) evaluation operator becomes the standard one. However, notice that schemes are not equivalent in the limit: scheme (19) tends to classic VI, while scheme (20) tends to Speedy Q-learning [6] (see the justification of the connection to SQL in Appx. B.2).

Next, we prove the equivalence for the case $\tau > 0$. We recall MD-MPI($\lambda$,$\tau$):

$$\begin{cases} \pi_{k+1} = \mathcal{G}_{\pi_k}^{\lambda;\tau}(q_k) \\ q_{k+1} = (T_{\pi_{k+1}|\pi_k})^m q_k + \epsilon_{k+1} \end{cases} . \tag{21}$$

Thanks to Eq. (5), we have that $\pi_{k+1} \propto \exp\frac{q_k + \lambda \ln \pi_k}{\lambda + \tau}$. We define $\beta = \frac{\lambda}{\lambda+\tau}$ (and thus $1 - \beta = \frac{\tau}{\lambda+\tau}$ and $\frac{\beta}{\lambda} = \frac{1}{\lambda+\tau}$). By induction, we have (writing "cst" any function depending solely on states, not necessarily the same for different lines):

$$\ln\pi_{k+1} = \frac{\beta}{\lambda}q_k + \beta\ln\pi_k + \mathrm{cst}$$

$$= \frac{\beta}{\lambda}\left(q_k + \beta q_{k-1} + \beta^2 q_{k-2} + \dots\right) + \mathrm{cst}$$

$$= \frac{\beta}{\lambda(1-\beta)}\left((1-\beta)(q_k + \beta q_{k-1} + \beta^2 q_{k-2} + \dots)\right) + \mathrm{cst}.$$

We now define $h_k$ as the moving average of past $q$-values, with $h_{-1} = 0$:

$$h_k = \beta h_{k-1} + (1-\beta)q_k = (1-\beta)\sum_{j=0}^{k}\beta^{k-j}q_j. \tag{22}$$

Noticing also that $\frac{\beta}{\lambda(1-\beta)} = \frac{1}{\tau}$, this shows that

$$\pi_{k+1} \propto \exp\frac{h_k}{\tau}.$$

As before, this means that $\pi_{k+1}$ is the solution of an entropy regularized greedy step with respect to $h_k$:

$$\pi_{k+1} = \underset{\pi \in \Delta_{\mathcal{A}}^{\mathcal{S}}}{\operatorname{argmax}} \left( \langle \pi, h_k \rangle + \tau \mathcal{H}(\pi) \right) = \mathcal{G}^{0,\tau}(h_k).$$

This means that Eq. (21) is equivalent to

$$\begin{cases} \pi_{k+1} = \mathcal{G}^{0,\tau}(h_k) \\ q_{k+1} = (T^{\lambda,\tau}_{\pi_{k+1}|\pi_k})^m q_k + \epsilon_{k+1} \\ h_{k+1} = \beta h_k + (1-\beta) q_{k+1} \text{ with } \beta = \frac{\lambda}{\lambda+\tau} \end{cases},$$

which is the DA-MPI($\lambda,\tau$) scheme. This concludes the proof.

## C.2 Proof of Theorem 1

Here, we provide the bound for DA-VI($\lambda$,0), which we recall:

$$\begin{cases} \pi_{k+1} = \mathcal{G}^{0,\frac{\lambda}{k+1}}(h_k) \\ q_{k+1} = T^{\lambda,0}_{\pi_{k+1}|\pi_k} q_k + \epsilon_{k+1} \\ h_{k+1} = \frac{k+1}{k+2} h_k + \frac{1}{k+2} q_{k+1} \end{cases}. \tag{23}$$

**Sketch of proof.** The quantity of interest is $q_* - q_{\pi_{k+1}}$, it can be decomposed as $q_* - q_{\pi_{k+1}} = q_* - h_k + h_k - q_{\pi_{k+1}}$. Lemma 1 allows expressing the quantity of interest essentially as a function of the Bellman residual $T_{\pi_{k+1}} h_k - h_k$. Controlling this residual is the key to state our bound. To achieve this, we first derive Lemma 2 that expresses the evaluation step (the update of the $q$-function) as a difference of Bellman operators applied to successive h-functions (the averages of $q$-values). Thanks to this, we're able to derive a Bellman-like recursion for $h_k$ in Lemma 3, using notably Lemma 2 and a telescoping argument. The rest of the proof consists in exploiting this Bellman-like recursion to control the residual and eventually boud the quantity of interest.

**Proof.** We start by stating a useful lemma.

**Lemma 1.** *For any $q \in \mathbb{R}^{\mathcal{S} \times \mathcal{A}}$ and $\pi \in \Delta_{\mathcal{A}}^{\mathcal{S}}$, we have*

$$q_\pi - q = (I - \gamma P_\pi)^{-1}(T_\pi q - q).$$

*Proof.* This result is classic, and appears many times in the literature (*e.g.*, Kakade and Langford [24]). We provide a one line proof for completeness, relying on basic properties of the Bellman operator:

$$q_\pi - q = T_\pi q_\pi - T_\pi q + T_\pi q - q = \gamma P_\pi(q_\pi - q) + T_\pi q - q \Leftrightarrow q_\pi - q = (I - \gamma P_\pi)^{-1}(T_\pi q - q).$$
$$\square$$

The aim is to bound the quantity $q_* - q_{\pi_{k+1}}$, the difference between the optimal value function and the value function computed by DA-VI($\lambda$,0). Thanks to Lemma 1, we can decompose this term as

$$\begin{aligned} q_* - q_{\pi_{k+1}} &= q_* - h_k + h_k - q_{\pi_{k+1}} \\ &= (I - \gamma P_{\pi_*})^{-1}(T_{\pi_*} h_k - h_k) - (I - \gamma P_{\pi_{k+1}})^{-1}(T_{\pi_{k+1}} h_k - h_k). \end{aligned} \tag{24}$$

Notice that $q_* = q_{\pi_*}$ for any optimal policy $\pi_*$. There exists an optimal deterministic policy [33], so we will consider a deterministic $\pi_*$. As for any deterministic policy, $\mathcal{H}(\pi_*) = 0$. Using the definition of $\pi_{k+1}$, we have

$$\pi_{k+1} = \mathcal{G}^{0,\frac{\lambda}{k+1}}(h_k) \Rightarrow \langle \pi_{k+1}, h_k \rangle + \frac{\lambda}{k+1} \mathcal{H}(\pi_{k+1}) \geq \langle \pi_*, h_k \rangle + \frac{\lambda}{k+1} \underbrace{\mathcal{H}(\pi_*)}_{=0}$$

$$\Rightarrow r + \gamma P \left( \langle \pi_{k+1}, h_k \rangle + \frac{\lambda}{k+1} \mathcal{H}(\pi_{k+1}) \right) \geq r + \gamma P \langle \pi_*, h_k \rangle$$

$$\Rightarrow T^{0,\frac{\lambda}{k+1}}_{\pi_{k+1}} h_k = T_{\pi_{k+1}} h_k + \gamma \frac{\lambda}{k+1} P \mathcal{H}(\pi_{k+1}) \geq T_{\pi_*} h_k.$$

Injecting this into Eq. (24), we obtain, using the fact that for any $\pi$ the matrix $(I - \gamma P_\pi)^{-1} = \sum_{t \geq 0} \gamma^t P_\pi^t$ is positive,

$$q_* - q_{\pi_{k+1}} \leq (I - \gamma P_{\pi_*})^{-1}(T_{\pi_{k+1}}^{0,\frac{\lambda}{k+1}} h_k - h_k) - (I - \gamma P_{\pi_{k+1}})^{-1}(T_{\pi_{k+1}}^{0,\frac{\lambda}{k+1}} h_k - h_k - \gamma \frac{\lambda}{k+1} P\mathcal{H}(\pi_{k+1})).$$
(25)

So, what we have to do is to control the residual $T_{\pi_{k+1}}^{0,\frac{\lambda}{k+1}} h_k - h_k$.

To do so, the following lemma will be useful.

**Lemma 2.** *For any $k \geq 1$, we have that*

$$T_{\pi_{k+1}|\pi_k}^{\lambda,0} q_k = (k+1) T_{\pi_{k+1}}^{0,\frac{\lambda}{k+1}} h_k - k T_{\pi_k}^{0,\frac{\lambda}{k}} h_{k-1}.$$

*For $k = 0$, we have*

$$T_{\pi_1|\pi_0}^{\lambda,0} q_0 = T_{\pi_1}^{0,\lambda} h_0 - \gamma \lambda P\mathcal{H}(\pi_0).$$

*Proof.* To prove this result, we will start by working on the optimization problem related to the regularized greedy step $\mathcal{G}_{\pi_k}^{\lambda,0} q_k$:

$$\langle \pi, q_k \rangle - \lambda \text{KL}(\pi \| \pi_k) = \langle \pi, q_k \rangle - \lambda \langle \pi, \ln \pi - \ln \pi_k \rangle = \langle \pi, q_k + \lambda \ln \pi_k \rangle - \lambda \langle \pi, \ln \pi \rangle.$$

For DA-VI$_1(\lambda,0)$, $\pi_{k+1} \in \mathcal{G}^{0,\frac{\lambda}{k+1}}(h_k)$ (see Eq. (23)), so according to Eq. (5), $\pi_{k+1} \propto \exp \frac{(k+1)h_k}{\lambda}$. Therefore, we have, using also the definition of $h_k$

$$q_k + \lambda \ln \pi_k = q_k + \lambda(\frac{k}{\lambda} h_{k-1} - \ln\langle 1, \exp \frac{kh_{k-1}}{\lambda} \rangle)$$

$$= (k+1)h_k - \lambda \ln\langle 1, \exp \frac{kh_{k-1}}{\lambda} \rangle.$$

Therefore, we have

$$\langle \pi, q_k \rangle - \lambda \text{KL}(\pi \| \pi_k) = \langle \pi, (k+1)h_k \rangle - \lambda \langle \pi, \ln \pi \rangle - \lambda \ln\langle \mathbf{1}, \exp \frac{kh_{k-1}}{\lambda} \rangle.$$

The maximizer is $\pi_{k+1}$, obviously. It is also the maximizer of $\langle \pi, (k+1)h_k \rangle - \lambda \langle \pi, \ln \pi \rangle$ (the third term not depending on $\pi$), and the associated maximum is, according to Eq. (4), $\lambda \ln\langle \mathbf{1}, \exp \frac{(k+1)h_k}{\lambda} \rangle$. This gives

$$\langle \pi_{k+1}, q_k \rangle - \lambda \text{KL}(\pi_{k+1} \| \pi_k) = \lambda \ln\langle \mathbf{1}, \exp \frac{(k+1)h_k}{\lambda} \rangle - \lambda \ln\langle \mathbf{1}, \exp \frac{kh_{k-1}}{\lambda} \rangle$$

$$= (k+1)\frac{\lambda}{k+1} \ln\langle \mathbf{1}, \exp \frac{(k+1)h_k}{\lambda} \rangle - k\frac{\lambda}{k} \ln\langle \mathbf{1}, \exp \frac{kh_{k-1}}{\lambda} \rangle.$$

Still from Eq. (4), we know that $\frac{\lambda}{k+1} \ln\langle 1, \exp \frac{(k+1)h_k}{\lambda} \rangle$ is the maximum of $\langle \pi, h_k \rangle + \frac{\lambda}{k+1}\mathcal{H}(\pi)$, the associated maximizer being again $\pi_{k+1}$, so using Eq. (6), we can conclude that

$$\langle \pi_{k+1}, q_k \rangle - \lambda \text{KL}(\pi_{k+1} \| \pi_k) = (k+1)\left(\langle \pi_{k+1}, h_k \rangle + \frac{\lambda}{k+1}\mathcal{H}(\pi_{k+1})\right) - k\left(\langle \pi_k, h_{k-1} \rangle + \frac{\lambda}{k}\mathcal{H}(\pi_k)\right).$$

Noticing that $r = (k+1)r - kr$, we have the first part of the result:

$$T_{\pi_{k+1}|\pi_k}^{\lambda,0} q_k = (k+1) T_{\pi_{k+1}}^{0,\frac{\lambda}{k+1}} h_k - k T_{\pi_k}^{0,\frac{\lambda}{k}} h_{k-1}.$$

This only holds for $k \geq 1$. For $k = 0$, using the fact that $h_0 = q_0$,

$$T_{\pi_1|\pi_0}^{\lambda,0} q_0 = r + \gamma P(\langle \pi_1, q_0 \rangle - \lambda \text{KL}(\pi_1 \| \pi_0))$$

$$= r + \gamma P(\langle \pi_1, h_0 \rangle - \lambda \langle \pi_1, \ln \pi_1 - \ln \pi_0 \rangle)$$

$$= r + \gamma P(\langle \pi_1, h_0 \rangle + \lambda \mathcal{H}(\pi_1) + \lambda \langle \pi_1, \ln \pi_0 \rangle)$$

$$= T_{\pi_1}^{0,\lambda} h_0 - \gamma \lambda P\mathcal{H}(\pi_0),$$

where we used in the last line the fact that, $\pi_0$ being uniform,

$$\langle \pi_1, \ln \pi_0 \rangle = \langle \pi_1, \ln \frac{1}{|\mathcal{A}|} \rangle = -\ln|\mathcal{A}|\langle \pi_1, 1 \rangle = -\ln|\mathcal{A}| = -\mathcal{H}(\pi_0).$$

This concludes the proof. $\square$

Using this lemma, we can provide a Bellman-like induction on $h_k$.

**Lemma 3.** *Define $E_k = -\sum_{j=1}^{k} \epsilon_j$. For any $k \geq 1$, we have that*

$$h_{k+1} = \frac{k+1}{k+2} T_{\pi_{k+1}}^{0,\frac{\lambda}{k+1}} h_k + \frac{1}{k+2} \left( q_0 - E_{k+1} - \gamma \lambda P \mathcal{H}(\pi_0) \right).$$

*Proof.* Using the definition of $h_k$, Lemma 2, the fact that $q_{k+1} = T_{\pi_{k+1}|\pi_k}^{\lambda,0} q_k + \epsilon_{k+1}$, and the definition $E_k = -\sum_{j=1}^{k} \epsilon_j$, we have

$$(k+2)h_{k+1} = \sum_{j=0}^{k+1} q_j$$

$$= q_0 + q_1 + \sum_{j=1}^{k} q_{j+1}$$

$$= q_0 + T_{\pi_1|\pi_0}^{\lambda,0} q_0 + \epsilon_1 + \sum_{j=1}^{k} \left( T_{\pi_{j+1}}^{\lambda,0} q_j + \epsilon_{j+1} \right)$$

$$= q_0 + T_{\pi_1}^{0,\lambda} h_0 - \gamma \lambda P \mathcal{H}(\pi_0) + \sum_{j=1}^{k} \left( (j+1) T_{\pi_{j+1}}^{0,\frac{\lambda}{j+1}} h_j - j T_{\pi_j}^{0,\frac{\lambda}{j}} h_{j-1} \right) - E_{k+1}$$

$$= q_0 - E_{k+1} - \gamma \lambda P \mathcal{H}(\pi_0) + (k+1) T_{\pi_{k+1}}^{0,\frac{\lambda}{k+1}} h_k$$

$$\Leftrightarrow h_{k+1} = \frac{k+1}{k+2} T_{\pi_{k+1}}^{0,\frac{\lambda}{k+1}} h_k + \frac{1}{k+2} \left( q_0 - E_{k+1} - \gamma \lambda P \mathcal{H}(\pi_0) \right).$$

$\square$

We now have the tools to work on the residual of interest. Starting from Lemma 3, and using the fact that $(k+2)h_{k+1} = (k+1)h_k + q_{k+1}$,

$$h_{k+1} = \frac{k+1}{k+2} T_{\pi_{k+1}}^{0,\frac{\lambda}{k+1}} h_k + \frac{1}{k+2} \left( q_0 - E_{k+1} - \gamma \lambda P \mathcal{H}(\pi_0) \right)$$

$$\Leftrightarrow (k+1)h_k + q_{k+1} = q_0 - E_{k+1} - \gamma \lambda P \mathcal{H}(\pi_0) + (k+1) T_{\pi_{k+1}}^{0,\frac{\lambda}{k+1}} h_k$$

$$\Leftrightarrow T_{\pi_{k+1}}^{0,\frac{\lambda}{k+1}} h_k - h_k = \frac{1}{k+1} \left( q_{k+1} - q_0 + E_{k+1} + \gamma \lambda P \mathcal{H}(\pi_0) \right).$$

Injecting this last result into decomposition (25), we get

$$q_* - q_{\pi_{k+1}} \leq (I - \gamma P_{\pi_*})^{-1} (T_{\pi_{k+1}}^{0,\frac{\lambda}{k+1}} h_k - h_k) - (I - \gamma P_{\pi_{k+1}})^{-1} (T_{\pi_{k+1}}^{0,\frac{\lambda}{k+1}} h_k - h_k - \gamma \lambda P \mathcal{H}(\pi_{k+1}))$$

$$= (I - \gamma P_{\pi_*})^{-1} \left( \frac{1}{k+1} (q_{k+1} - q_0 + E_{k+1} + \gamma \lambda P \mathcal{H}(\pi_0)) \right)$$

$$- (I - \gamma P_{\pi_{k+1}})^{-1} \left( \frac{1}{k+1} (q_{k+1} - q_0 + E_{k+1} + \gamma \lambda P \mathcal{H}(\pi_0)) - \gamma \frac{\lambda}{k+1} P \mathcal{H}(\pi_{k+1}) \right)$$

$$\leq (I - \gamma P_{\pi_*})^{-1} \left( \frac{1}{k+1} (q_{k+1} - q_0 + E_{k+1} + \gamma \lambda P \mathcal{H}(\pi_0)) \right)$$

$$- (I - \gamma P_{\pi_{k+1}})^{-1} \left( \frac{1}{k+1} (q_{k+1} - q_0 + E_{k+1} - \gamma \lambda P \mathcal{H}(\pi_{k+1})) \right),$$

where we used for the last inequality the fact that $-(I - \gamma P_{\pi_{k+1}})^{-1} P\mathcal{H}(\pi_0) \leq 0$. Next, using the fact that $q_* - q_{\pi_{k+1}} \geq 0$ and rearranging terms, we have

$$q_* - q_{\pi_{k+1}} \leq \left| \left( (I - \gamma P_{\pi_*})^{-1} - (I - \gamma P_{\pi_{k+1}})^{-1} \right) \frac{E_{k+1}}{k+1} \right|$$

$$+ (I - \gamma P_{\pi_*})^{-1} \left| \frac{q_{k+1} - q_0 + \gamma\lambda P\mathcal{H}(\pi_0)}{k+1} \right|$$

$$+ (I - \gamma P_{\pi_{k+1}})^{-1} \left| \frac{q_{k+1} - q_0 + \gamma\lambda P\mathcal{H}(\pi_{k+1})}{k+1} \right|.$$

We assumed that $\|q_{k+1}\|_\infty \leq v_{\max} \leq v_{\max}^\lambda$ (see also Rk. 1). When introducing the algorithm, we assumed that $\|q_0\|_\infty \leq v_{\max}$. Therefore, $\|q_0 - \gamma\lambda P\mathcal{H}(\pi_0)\|_\infty \leq v_{\max}^\lambda$. Writing $\mathbf{1}$ the vector whose components are all 1, we get $|q_{k+1} - q_0 + \gamma\lambda P\mathcal{H}(\pi_0)| \leq 2v_{\max}^\lambda \mathbf{1}$. Notice that for any policy $\pi$, we have that $P_\pi \mathbf{1} = \mathbf{1}$. Therefore, we have

$$(I - \gamma P_{\pi_*})^{-1} \left| \frac{q_{k+1} - q_0 + \gamma\lambda P\mathcal{H}(\pi_0)}{k+1} \right| \leq \frac{2}{1-\gamma} \frac{v_{\max}^\lambda}{k+1} \mathbf{1}.$$

With the same arguments, we have that

$$(I - \gamma P_{\pi_{k+1}})^{-1} \left| \frac{q_{k+1} - q_0 + \gamma\lambda P\mathcal{H}(\pi_{k+1})}{k+1} \right| \leq \frac{2}{1-\gamma} \frac{v_{\max}^\lambda}{k+1} \mathbf{1}.$$

We finally have

$$q_* - q_{\pi_{k+1}} \leq \left| \left( (I - \gamma P_{\pi_*})^{-1} - (I - \gamma P_{\pi_{k+1}})^{-1} \right) \frac{E_{k+1}}{k+1} \right| + \frac{4}{1-\gamma} \frac{v_{\max}^\lambda}{k+1} \mathbf{1},$$

which is the stated result.

### C.3  About Remark 1

We stated in Rk. 1, in the context of DA-VI($\lambda$,0), that the assumption $\|q_k\|_\infty \leq v_{\max}$ is not strong with approximation, as this just requires clipping the $q$-values. Indeed, without approximation, it's not even necessary to clip the $q$-values.

**No approximation.**  We will proceed by induction. Assume that $\|q_k\|_\infty \leq v_{\max}$. We assumed generally that $\|q_0\|_\infty \leq v_{\max}$. Without error, the considered scheme is

$$\begin{cases} \pi_{k+1} = \mathcal{G}^{0,\frac{\lambda}{k+1}}(h_k) \\ q_{k+1} = T^{\lambda,0}_{\pi_{k+1}|\pi_k} q_k \\ h_{k+1} = \frac{k+1}{k+2} h_k + \frac{1}{k+2} q_{k+1} \end{cases} \Leftrightarrow \begin{cases} \pi_{k+1} = \mathcal{G}^{\lambda,0}(q_k) \\ q_{k+1} = T^{\lambda,0}_{\pi_{k+1}|\pi_k} q_k \end{cases} .$$

As $\pi_{k+1} = \mathcal{G}^{\lambda,0}(q_k)$, we have that

$$q_{k+1} = T^{\lambda,0}_{\pi_{k+1}|\pi_k} q_k \geq T^{\lambda,0}_{\pi_k|\pi_k} q_k = T_{\pi_k} q_k \geq -v_{\max}\mathbf{1},$$

The inequality making use of the induction argument. On the other hand, making use of the positiveness of the KL divergence, we have that

$$q_{k+1} = T^{\lambda,0}_{\pi_{k+1}|\pi_k} q_k \leq T_{\pi_{k+1}} q_k \leq v_{\max}\mathbf{1},$$

where again the inequality comes from the induction argument. This allows concluding, $\|q_{k+1}\|_\infty \leq v_{\max}$.

**With approximation.**  Knowing a bound of the $q$-values without approximation, we can clip $q_k$ such that it satisfies the bound, the effect of the clipping being part of the error. For example, assume that the evaluation step is approximated with a least-squares problems, a parameterized $q$-function, the target being a sampling of $T^{\lambda,0}_{\pi_{k+1}|\pi_k} q_k$, $q_k$ being the previous approximation (for example the target network). We can clip the result of the least-squares in $[-v_{\max}, +v_{\max}]$ and call the resulting function $q_{k+1}$. The resulting error is defined as $\epsilon_{k+1} = q_{k+1} - T^{\lambda,0}_{\pi_{k+1}|\pi_k} q_k$.

## C.4 Proof of Theorem 2

In this section, we provide a bound for DA-VI($\lambda$,$\tau$). First, we recall the scheme:

$$\begin{cases} \pi_{k+1} = \mathcal{G}^{0,\tau}(h_k) \\ q_{k+1} = T^{\lambda,\tau}_{\pi_{k+1}|\pi_k} q_k + \epsilon_{k+1} \\ h_{k+1} = \beta h_k + (1-\beta)q_{k+1} \text{ with } \beta = \frac{\lambda}{\lambda+\tau} \end{cases}.$$

We recall that due to the entropy term, this scheme cannot converge to the unregularized optimal $q_*$ function. Yet, without errors and with $\lambda = 0$, it would converge to the solution of the MDP regularized by the scaled entropy [20] (optimizing for the reward augmented by the scaled entropy). Our bound will show that adding a KL penalty does not change this. We recall the notations introduced in the main paper. We already have defined the operator $T^{0,\tau}_\pi$. It has a unique fixed point, which we write $q^\tau_\pi$. The unique optimal $q$-function is $q^\tau_* = \max_\pi q^\tau_\pi$. We write $\pi^\tau_* = \mathcal{G}^{0,\tau}(q^\tau_*)$ the associated unique optimal policy, and $q^\tau_{\pi^\tau_*} = q^\tau_*$.

**Sketch of proof.** The proof is similar to the one of Thm. 1, albeit a bit more technical. Thanks to Lemma 4 (that generalizes Lemma 1), we decompose the quantity of interest $q^\tau_* - q^\tau_{\pi_{k+1}}$ as a function of $q^\tau_* - h_k$ and of $T^{0,\tau}_{\pi_{k+1}} h_k - h_k$, to be respectively upper-bounded and lower-bounded. To achieve this, we first derive Lemma 5 that expresses the evaluation step as a difference of Bellman operators applied to successive h-functions (similarly to Lemma 2). Thanks to this, we're able to derive a Bellman-like recursion for $h_k$ in Lemma 6, using notably Lemma 5 and a telescoping argument (similarly to Lemma 3). The end of the proof is then close to the classic propagation of errors of AVI, involving moving averages of the errors instead of the errors, as well as some additional terms.

**Proof.** The following lemma, generalizing Lemma 1 to the regularized Bellman operator, will be useful:

**Lemma 4.** *Let $\tau \geq 0$. For any $q \in \mathbb{R}^{\mathcal{S}\times\mathcal{A}}$ and $\pi \in \Delta^{\mathcal{S}}_{\mathcal{A}}$, we have*

$$q^\tau_\pi - q = (I - \gamma P_\pi)^{-1}(T^{0,\tau}_\pi q - q).$$

*Proof.* The proof is the same as the one of Lemma 1, relying on the fact that the regularized Bellman operator has the same properies as the Bellman operator [20]:

$$q^\tau_\pi - q = T^{0,\tau}_\pi q^\tau_\pi - T^{0,\tau}_\pi q + T^{0,\tau}_\pi q - q = \gamma P_\pi(q^\tau_\pi - q) + T^{0,\tau}_\pi q - q \Leftrightarrow q^\tau_\pi - q = (I - \gamma P_\pi)^{-1}(T^{0,\tau}_\pi q - q).$$

$\square$

We will bound the quantity $q^\tau_* - q^\tau_{\pi_{k+1}}$, using the following decomposition, based on Lemma 4:

$$\begin{aligned} q^\tau_* - q^\tau_{\pi_{k+1}} &= q^\tau_* - h_k + h_k - q^\tau_{\pi_{k+1}} \\ &= (q^\tau_* - h_k) - (I - \gamma P_{\pi_{k+1}})^{-1}(T^{0,\tau}_{\pi_{k+1}} h_k - h_k). \end{aligned} \quad (26)$$

To do so, we will upper-bound $q^\tau_* - h_k$ and lower-bound $T^{0,\tau}_{\pi_{k+1}} h_k - h_k$ (we recall that the matrix $(I - \gamma P_{\pi_{k+1}})^{-1}$ is non-negative). This requires a Bellman-like induction on $h_k$. For this, the following intermediate lemma, similar to Lemma 2, will be useful.

**Lemma 5.** *For any $k \geq 0$, we have that*

$$T^{\lambda,\tau}_{\pi_{k+1}|\pi_k} q_k = \frac{1}{1-\beta}\left(T^{0,\tau}_{\pi_{k+1}} h_k - \beta T^{0,\tau}_{\pi_k} h_{k-1}\right).$$

*Proof.* We have that, for any $\pi$,

$$\begin{aligned} \langle \pi, q_k \rangle - \lambda \,\mathrm{KL}(\pi\|\pi_k) + \tau\mathcal{H}(\pi) &= \langle \pi, q_k \rangle - \lambda\langle\pi, \ln\pi - \ln\pi_k\rangle - \tau\langle\pi, \ln\pi\rangle \\ &= \langle\pi, q_k + \lambda\ln\pi_k\rangle - (\lambda+\tau)\langle\pi, \ln\pi\rangle. \end{aligned}$$

As $\pi_{k+1} \propto \exp \frac{h_k}{\tau}$, using also the fact that $\beta = \frac{\lambda}{\lambda + \tau}$ and $1 - \beta = \frac{\tau}{\lambda + \tau}$, as well as the definition of $h_k$ (22), we have

$$q_k + \lambda \ln \pi_k = q_k + \lambda \left( \frac{h_{k-1}}{\tau} - \ln \langle \mathbf{1}, \exp \frac{h_{k-1}}{\tau} \rangle \right)$$

$$= \frac{1}{1 - \beta} \left( (1 - \beta)q_k + \beta h_{k-1} - \beta \tau \ln \langle \mathbf{1}, \exp \frac{h_{k-1}}{\tau} \rangle \right)$$

$$= \frac{1}{1 - \beta} \left( h_k - \beta \tau \ln \langle \mathbf{1}, \exp \frac{h_{k-1}}{\tau} \rangle \right).$$

Hence, injecting this in the previous result, we get

$$\langle \pi, q_k + \lambda \ln \pi_k \rangle - (\lambda + \tau) \langle \pi, \ln \pi \rangle = \langle \pi, q_k + \lambda \ln \pi_k \rangle - \frac{\tau}{1 - \beta} \langle \pi, \ln \pi \rangle$$

$$= \frac{1}{1 - \beta} \left( \langle \pi, h_k \rangle - \tau \langle \pi, \ln \pi \rangle - \beta \tau \ln \langle \mathbf{1}, \exp \frac{h_{k-1}}{\tau} \rangle \right).$$

Now, as $\pi_{k+1} \propto \exp \frac{h_k}{\tau}$, we have that $\langle \pi_{k+1}, h_k \rangle + \tau \mathcal{H}(\pi_{k+1}) = \tau \ln \langle \mathbf{1}, \exp \frac{h_k}{\tau} \rangle$ (again from Eq. (6)), therefore

$$\langle \pi_{k+1}, q_k \rangle - \lambda \operatorname{KL}(\pi_{k+1} \| \pi_k) + \tau \mathcal{H}(\pi_{k+1})$$

$$= \frac{1}{1 - \beta} \left( \langle \pi_{k+1}, h_k \rangle + \tau \mathcal{H}(\pi_{k+1}) - \beta (\langle \pi_k, h_{k-1} \rangle + \tau \mathcal{H}(\pi_k)) \right).$$

The result follows by the definition of $T^{\lambda, \tau}_{\pi_{k+1} | \pi_k} q_k = r + \gamma P(\langle \pi_{k+1}, q_k \rangle - \lambda \operatorname{KL}(\pi_{k+1} \| \pi_k) + \tau \mathcal{H}(\pi_{k+1}))$, and noticing that $r = \frac{1}{1-\beta}(r - \beta r)$. $\qquad \square$

This result allows to build the lemma stating a Bellman-like induction for $h_k$.

**Lemma 6.** *Define* $E^\beta_{k+1} = -(1 - \beta) \sum_{j=1}^{k+1} \beta^{k+1-j} \epsilon_j = \beta E^\beta_k + (1 - \beta) \epsilon_{k+1}$ *(with* $E^\beta_0 = 0$*).*
*For any* $k \geq 0$*, we have that*

$$h_{k+1} = T^{0, \tau}_{\pi_{k+1}} h_k - E_{k+1} - \beta^{k+1} (T^{0, \tau}_{\pi_0} h_{-1} - h_0).$$

*Proof.* Using the definition of $h_k$, Eq. (22), the relationship between $q_{k+1}$ and $q_k$, and Lemma 5, we have

$$h_{k+1} = (1 - \beta) \sum_{j=0}^{k+1} \beta^{k+1-j} q_k$$

$$= (1 - \beta) \beta^{k+1} q_0 + (1 - \beta) \sum_{j=1}^{k+1} \beta^{k+1-j} q_j$$

$$= (1 - \beta) \beta^{k+1} q_0 + (1 - \beta) \sum_{j=0}^{k} \beta^{k-j} q_{j+1}$$

$$= (1 - \beta) \beta^{k+1} q_0 + (1 - \beta) \sum_{j=0}^{k} \beta^{k-j} \left( T^{\lambda, \tau}_{\pi_{j+1} | \pi_j} q_j + \epsilon_{j+1} \right)$$

$$= (1 - \beta) \beta^{k+1} q_0 + (1 - \beta) \sum_{j=0}^{k} \beta^{k-j} \left( \frac{1}{1 - \beta} \left( T^{0, \tau}_{\pi_{j+1}} h_j - \beta T^{0, \tau}_{\pi_j} h_{j-1} \right) + \epsilon_{j+1} \right).$$

Let define $E^{\beta}_{k+1}$ as

$$E_{k+1} = -(1-\beta)\sum_{j=0}^{k}\beta^{k-j}\epsilon_{j+1}$$

$$= -(1-\beta)\sum_{j=1}^{k+1}\beta^{k+1-j}\epsilon_j$$

$$= \beta E^{\beta}_k + (1-\beta)\epsilon_{k+1} \text{ with } E_0 = 0.$$

We also have

$$(1-\beta)\sum_{j=0}^{k}\beta^{k-j}\left(\frac{1}{1-\beta}\left(T^{0,\tau}_{\pi_{j+1}}h_j - \beta T^{0,\tau}_{\pi_j}h_{j-1}\right)\right)$$

$$=\sum_{j=0}^{k}\beta^{k-j}\left(T^{0,\tau}_{\pi_{j+1}}h_j - \beta T^{0,\tau}_{\pi_j}h_{j-1}\right)$$

$$=\sum_{j=1}^{k+1}\beta^{k+1-j}T^{0,\tau}_{\pi_j}h_{j-1} - \sum_{j=0}^{k}\beta^{k+1-j}T^{0,\tau}_{\pi_j}h_{j-1}$$

$$=T^{0,\tau}_{\pi_{k+1}}h_k - \beta^{k+1}T^{0,\tau}_{\pi_0}h_{-1}.$$

Notice also that $h_0 = (1-\beta)q_0$. Putting all these parts together, we obtain

$$h_{k+1} = \beta^{k+1}h_0 - E^{\beta}_{k+1} + T^{0,\tau}_{\pi_{k+1}}h_k - \beta^{k+1}T^{0,\tau}_{\pi_0}h_{-1}$$

$$= T^{0,\tau}_{\pi_{k+1}}h_k - E^{\beta}_{k+1} - \beta^{k+1}(T^{0,\tau}_{\pi_0}h_{-1} - h_0),$$

which is the stated result. $\qquad\square$

Thanks to this result, we can now bound the terms of interest.

**Upper-bounding $q^{\tau}_* - h_k$.** Write $e_k = E^{\beta}_k + \beta^k(T^{0,\tau}_{\pi_0}h_{-1} - h_0)$, we have from Lemma 6 that $h_{k+1} = T^{0,\tau}_{\pi_{k+1}}h_k - e_{k+1}$. Then, we have :

$$q^{\tau}_* - h_{k+1} = q^{\tau}_* - T^{0,\tau}_{\pi_{k+1}}h_k + e_{k+1}$$

$$= \underbrace{T^{0,\tau}_{\pi^{\tau}_*}q^{\tau}_* - T^{0,\tau}_{\pi^{\tau}_*}h_k}_{=\gamma P_{\pi^{\tau}_*}(q^{\tau}_* - h_k)} + \underbrace{T^{0,\tau}_{\pi^{\tau}_*}h_k - T^{0,\tau}_{\pi_{k+1}}h_k}_{\leq 0 \text{ as } \pi_{k+1}=\mathcal{G}^{0,\tau}(h_k)} + e_{k+1}$$

$$\leq \gamma P_{\pi^{\tau}_*}(q^{\tau}_* - h_k) + e_{k+1}.$$

By direct induction, we obtain

$$q^{\tau}_* - h_{k+1} \leq (\gamma P_{\pi^{\tau}_*})^{k+1}(q^{\tau}_* - h_0) + \sum_{j=1}^{k+1}(\gamma P_{\pi^{\tau}_*})^{k+1-j}e_j$$

$$= (\gamma P_{\pi^{\tau}_*})^{k+1}(q^{\tau}_* - h_0) + \sum_{j=1}^{k+1}(\gamma P_{\pi^{\tau}_*})^{k+1-j}\left(E^{\beta}_j + \beta^j(T^{0,\tau}_{\pi_0}h_{-1} - h_0)\right). \quad (27)$$

This is the desired upper-bound.

**Lower-bounding $T^{0,\tau}_{\pi_{k+1}}h_k - h_k$.** Using the same notation $e_k$, we have

$$T^{0,\tau}_{\pi_{k+1}}h_k - h_k = \underbrace{T^{0,\tau}_{\pi_{k+1}}h_k - T^{0,\tau}_{\pi_k}h_k}_{\geq 0 \text{ as } \pi_{k+1}=\mathcal{G}^{0,\tau}(h_k)} + T^{0,\tau}_{\pi_k}h_k - h_k$$

$$\geq T^{0,\tau}_{\pi_k}h_k - h_k$$

$$= T^{0,\tau}_{\pi_k}\left(T^{0,\tau}_{\pi_k}h_{k-1} - e_k\right) - \left(T^{0,\tau}_{\pi_k}h_{k-1} - e_k\right) \text{ by Lemma 6}$$

$$= \gamma P_{\pi_k}\left(T^{0,\tau}_{\pi_k}h_{k-1} - h_{k-1}\right) - (I - \gamma P_{\pi_k})^{-1}e_k.$$

We define $P_{k:j} = P_{\pi_k} P_{\pi_{k-1}} \ldots P_{\pi_{j+1}} P_{\pi_j}$ for $j \leq k$, with the convention $P_{k:k+1} = I$. By direct induction, the preceding inequality gives

$$
T_{\pi_{k+1}}^{0,\tau} h_k - h_k \geq \gamma^k P_{k:1}(T_{\pi_1}^{0,\tau} h_0 - h_0) - \sum_{j=1}^{k} \gamma^{k-j} P_{k:j+1}(I - \gamma P_{\pi_j}) e_j
$$

$$
= \gamma^k P_{k:1}(T_{\pi_1}^{0,\tau} h_0 - h_0) - \sum_{j=1}^{k} \gamma^{k-j} P_{k:j+1}(I - \gamma P_{\pi_j})(E_j^\beta + \beta^j(T_{\pi_0}^{0,\tau} h_{-1} - h_0)).
$$
(28)

**Putting things together.** Plugging Eqs. (27) and (28) into Eq. (26), we obtain

$$
q_*^\tau - q_{\pi_{k+1}}^\tau \leq (\gamma P_{\pi_*^\tau})^k (q_*^\tau - h_0) + \sum_{j=1}^{k} (\gamma P_{\pi_*^\tau})^{k-j} \left( E_j^\beta + \beta^j (T_{\pi_0}^{0,\tau} h_{-1} - h_0) \right)
$$

$$
+ (I - \gamma P_{\pi_{k+1}})^{-1} \Bigg( - \gamma^k P_{k:1}(T_{\pi_1}^{0,\tau} h_0 - h_0)
$$

$$
+ \sum_{j=1}^{k} \gamma^{k-j} P_{k:j+1}(I - \gamma P_{\pi_j})(E_j^\beta + \beta^j (T_{\pi_0}^{0,\tau} h_{-1} - h_0)) \Bigg).
$$

Using the fact that $q_*^\tau - q_{\pi_{k+1}}^\tau \geq 0$, rearranging terms, we have

$$
q_*^\tau - q_{\pi_{k+1}}^\tau \leq \sum_{j=1}^{k} \left| (\gamma P_{\pi_*^\tau})^{k-j} + (I - \gamma P_{\pi_{k+1}})^{-1} \gamma^{k-j} P_{k:j+1} \left( I - \gamma P_{\pi_j} \right) E_j^\beta \right|
$$

$$
+ (\gamma P_{\pi_*^\tau})^k |q_*^\tau - h_0| + \sum_{j=1}^{k} (\gamma P_{\pi_*^\tau})^{k-j} \beta^j |T_{\pi_0}^{0,\tau} h_{-1} - h_0|
$$

$$
+ (I - \gamma P_{\pi_{k+1}})^{-1} \gamma^k P_{k:1} |T_{\pi_1}^{0,\tau} h_0 - h_0|
$$

$$
+ (I - \gamma P_{\pi_{k+1}})^{-1} \sum_{j=1}^{k} \gamma^{k-j} P_{k:j+1}(I + \gamma P_{\pi_j}) \beta^j |T_{\pi_0}^{0,\tau} h_{-1} - h_0|.
$$
(29)

The first term is related to the error, the others to the initialisation. We'll work on each of these other terms.

Recall that we assumed that $\|q_0\|_\infty \leq v_{\max} = \frac{r_{\max}}{1-\gamma}$. Therefore, $\|q_0\|_\infty \leq v_{\max}^\tau = \frac{r_{\max} + \tau \ln |\mathcal{A}|}{1-\gamma}$. As $h_0 = (1-\beta)q_0$, we have $\|h_0\|_\infty \leq (1-\beta)v_{\max}^\tau$. From obvious properties of regularized MDPs [20], we have $\|q_*^\tau\|_\infty \leq v_{\max}^\tau$. Therefore, writing $\mathbf{1} \in \mathbb{R}^{S \times A}$ the vector with all components equal to 1, we have $|q_*^\tau - h_0| \leq (2-\beta)v_{\max}^\tau \mathbf{1}$. Notice that for any policy $\pi$, we have $P_\pi \mathbf{1} = \mathbf{1}$, thus

$$
(\gamma P_{\pi_*^\tau})^k |q_*^\tau - h_0| \leq \gamma^k (2-\beta) v_{\max}^\tau \mathbf{1}.
$$

We also have that $\|T_{\pi_1}^{0,\tau} h_0\|_\infty \leq r_{\max} + \tau \ln |\mathcal{A}| + \gamma(1-\beta)v_{\max}^\tau = (1 - \gamma\beta)v_{\max}^\tau$, so

$$
(I - \gamma P_{\pi_{k+1}})^{-1} \gamma^k P_{k:1} |T_{\pi_1}^{0,\tau} h_0 - h_0| \leq \gamma^k \frac{2 - (1+\gamma)\beta}{1-\gamma} v_{\max}^\tau \mathbf{1}.
$$

By definition $h_{-1} = 0$, so we have $\|T_{\pi_0}^{0,\tau} h_{-1}\|_\infty = \|r + \gamma P \tau \mathcal{H}(\pi_0)\|_\infty \leq r_{\max} + \tau \ln |\mathcal{A}| = (1-\gamma)v_{\max}^\tau$, so $\|T_{\pi_0}^{0,\tau} h_{-1} - h_0\|_\infty \leq (2 - \gamma - \beta)v_{\max}^\tau$. Therefore, we have the following bound:

$$
\sum_{j=1}^{k} (\gamma P_{\pi_*^\tau})^{k-j} \beta^j |T_{\pi_0}^{0,\tau} h_{-1} - h_0| \leq \gamma^k \sum_{j=1}^{k} \left( \frac{\beta}{\gamma} \right)^j (2 - \beta - \gamma) v_{\max}^\tau \mathbf{1}.
$$

Similarly, for the last term we have

$$
(I - \gamma P_{\pi_{k+1}})^{-1} \sum_{j=1}^{k} \gamma^{k-j} P_{k:j+1}(I + \gamma P_{\pi_j}) \beta^j |T_{\pi_0}^{0,\tau} h_{-1} - h_0| \leq \frac{1+\gamma}{1-\gamma} \gamma^k \sum_{j=1}^{k} \left( \frac{\beta}{\gamma} \right)^j (2 - \beta - \gamma) v_{\max}^\tau \mathbf{1}.
$$

Summing these four upper bounds, we obtain

$$\gamma^k(2-\beta)v_{\max}^\tau\mathbf{1} + \gamma^k\frac{2-(1+\gamma)\beta}{1-\gamma}v_{\max}^\tau\mathbf{1} + \gamma^k\sum_{j=1}^{k}\left(\frac{\beta}{\gamma}\right)^j(2-\beta-\gamma)v_{\max}^\tau\mathbf{1}$$

$$+\frac{1+\gamma}{1-\gamma}\gamma^k\sum_{j=1}^{k}\left(\frac{\beta}{\gamma}\right)^j(2-\beta-\gamma)v_{\max}^\tau\mathbf{1}$$

$$=2\gamma^k\frac{2-\beta-\gamma}{1-\gamma}\sum_{j=0}^{k}\left(\frac{\beta}{\gamma}\right)^j v_{\max}^\tau\mathbf{1} = 2\gamma^k\left(1+\frac{1-\beta}{1-\gamma}\right)\sum_{j=0}^{k}\left(\frac{\beta}{\gamma}\right)^j v_{\max}^\tau\mathbf{1}.$$

Plugging this result into Eq. (29), we obtain the stated result:

$$q_*^\tau - q_{\pi_{k+1}}^\tau \le \sum_{j=1}^{k}\left|(\gamma P_{\pi_*^\tau})^{k-j} + (I-\gamma P_{\pi_{k+1}})^{-1}\gamma^{k-j}P_{k:j+1}\left(I-\gamma P_{\pi_j}\right)E_j^\beta\right|$$

$$+ \gamma^k\left(1+\frac{1-\beta}{1-\gamma}\right)\sum_{j=0}^{k}\left(\frac{\beta}{\gamma}\right)^j v_{\max}^\tau\mathbf{1}.$$

## D   Empirical illustration of the bounds

We have illustrated the bounds of Sec. 2 (Fig. 1) in a simple tabular setting with acces to a generative model. We provide more details about this setting here.

We consider MDPs with small state and action spaces, such that a tabular representation of the $q$-function is possible. We also assume to have access to a generative model, allowing us to sample a transition for any state-action couple. We then consider sampled MD-VI($\lambda,\tau$), depicted in Alg. 1. At each iteration of MD-VI, we sample a single transition for each state-action couple and apply the resulting sampled Bellman operator. The error $\epsilon_k$ is the difference between the sampled and the exact operators. The sequence of these estimation errors is thus a martingale difference w.r.t. its natural filtration [6] (one can think about bounded, centered and roughly i.i.d. errors).

We run this algorithm on randomized MDPs called Garnets. A Garnet [4] is an abstract MDP, built from three parameters $(N_S, N_A, N_B)$, with $N_S$ and $N_A$ respectively the number of states and actions, and $N_B$ the branching factor. The principle is to directly build the transition kernel $P$ that represents the MDP. For each $(s,a) \in \mathcal{S} \times \mathcal{A}$, $N_B$ states $(s_1, \ldots s_{N_B})$ are drawn uniformly from $\mathcal{S}$ without replacement. Then, $N_B-1$ numbers are drawn uniformly in $(0,1)$ and sorted as $(p_0 = 0, p_1, \ldots p_{N_B-1}, p_{N_B} = 1)$. The transition kernel is then defined as $P(s_k|s,a) = p_k - p_{k-1}$ for each $1 \le k \le N_B$. The reward function is drawn uniformly in $(0,1)$ for 10% of the states, these states being drawn uniformly without replacement.

For the experiments shown in Fig. 1, we set $N_S = 30$, $N_A = 4$, $N_B = 4$ and $\gamma = 0.9$. We generate 100 Garnets and run MD-VI once for each of these Garnets, for $K = 800$ iterations. The results in Fig. 1 shows the normalized average performance, $\frac{\|q_*^\tau - q_{\pi_k}^\tau\|_1}{\|q_*^\tau\|_1}$. For sampled DA-VI($\lambda, 0$), we show the behavior for various values of $\lambda$. For DA-VI($\lambda, \tau$), we fix $\tau$ to a small value ($\tau = 10^{-3}$) and show the behavior for various values of $\beta = \frac{\lambda}{\lambda+\tau}$. Notice that considering a large value of $\tau$ would not be interesting. In this case, the regularized optimal policy would be close to be uniform, so close to the initial policy.

## E   Algorithms and experimental details

This appendix provides additional details about the algorithms and the experiments:

- Appx. E.1 provides a complementary high level view of algorithms sketched in Sec. 5.
- Appx. E.2 provides implementation details of these algorithms, including a pseudo-code.

---

**Algorithm 1** Sampled MD-VI($\lambda, \tau$)

---

**Require:** $K$ number of iterations, $P$ the transition kernel.
**set** $\beta = \frac{\lambda}{\lambda + \tau}$
**set** $q_0$ to the null vector
**set** $\pi_0$ to be the uniform policy
  **for** $1 \leq k \leq K$ **do**
    **for** $(s, a) \in \mathcal{S} \times \mathcal{A}$ **do**

$$\pi_k(a|s) = \frac{\pi_{k-1}(a|s)^\beta \exp \frac{q_{k-1}(s,a)}{\lambda + \tau}}{\sum_{b \in \mathcal{A}} \pi_{k-1}(b|s)^\beta \exp \frac{q_{k-1}(s,b)}{\lambda + \tau}}$$

    **end for**
    **for** $(s, a) \in \mathcal{S} \times \mathcal{A}$ **do**
    $s' \sim P(\cdot|s,a)$

$$q_k(s,a) = r(s,a) + \gamma \sum_{b \in \mathcal{A}} \pi_k(b|s') \left( q_{k-1}(s',b) - \lambda \ln \frac{\pi_k(b|s')}{\pi_{k-1}(b|s')} - \tau \ln \pi_k(b|s') \right)$$

    **end for**
  **end for**
**output** $\pi_K$

---

- Appx. E.3 provides all hyperparameters used in our experiments.
- Appx. E.4 provides additionnal experiments (one additional gym environment, Lunar Lander, and two additional Atari games, Breakout and Seaquest), as well as additional visualisations (including all training curves on Atari games).

### E.1 High level view of practical algorithms

DA-VI and MD-VI are extensions of VI. One of the most prevalent VI-based deep RL algorithm is probably DQN [27]. Thus, our approach consists in modifying the DQN algorithm to study regularization. To complement the sketch of Sec. 5, We present the different variations we consider with a high level viewpoint here, all practical details being just after.

DQN maintains a replay buffer and a target network $q_k$, and computes $q_{k+1}$ by minimizing the loss (recall that '$w/o$' stands for "without regularization"):

$$\mathcal{L}_{w/o}(q) = \hat{\mathbb{E}}_{s,a} \left[ \left( [\hat{T}_{\pi_{k+1}} q_k](s,a) - q(s,a) \right)^2 \right], \tag{30}$$

with $q$ a neural network, $\pi_{k+1} \in \mathcal{G}(q_k)$ the greedy policy computed analytically from $q_k$, $[\hat{T}_{\pi_{k+1}} q_k](s,a) = r(s,a) + \gamma \langle \pi_{k+1}, q_k \rangle (s')$ the sampled Bellman operator (with $s' \sim P(\cdot|s,a)$), and where the empirical expectation $\hat{\mathbb{E}}_{s,a}$ is according to the transitions in the buffer. DQN is an optimistic AVI scheme, in the sense that only a few steps of stochastic gradient descent are performed before updating the target network. We modify DQN by adding a policy network and possibly modifying the evaluation step. For the moment, we consider $\tau > 0$.

**Greedy step.** As explained before, when the greedy step is approximated, MD-VI and DA-VI are no longer equivalent. We start with MD-VI. A natural way to learn the policy network is to optimize directly for the greedy step. Let $\pi_k$ be the target policy network and $q_k$ the target $q$-network, it corresponds to ('dir' stands for direct):

$$\mathcal{L}_{\mathrm{dir}}(\pi) = \hat{\mathbb{E}}_s \left[ \langle \pi, q_k \rangle(s) - \lambda \, \mathrm{KL}(\pi || \pi_k)(s) + \tau \mathcal{H}(\pi)(s) \right]. \tag{31}$$

Maximizing this loss over networks gives $\pi_{k+1}$. This is reminiscent of TRPO (see Appx. B.1).

One can also compute analytically the policy $\pi_{k+1}$ (see Appx. A), but it would require remembering all past networks. Thus, another solution is to approximate this analytical solution by a neural network ('ind' stands for indirect):

$$\mathcal{L}_{\mathrm{ind}}(\pi) = \hat{\mathbb{E}}_s \left[ \mathrm{KL}(\pi_{k+1}^* || \pi)(s) \right] \text{ with } \pi_{k+1}^* \propto \pi_k^\beta \exp \frac{\beta q_k}{\lambda}.$$

Minimizing this loss over networks gives $\pi_{k+1}$. This is reminiscent of MPO (see Appx. B.1), up to the fact that we consider the KL in the reverse order. Indeed, MPO (or SAC) would optimise for $\hat{\mathbb{E}}_s[\mathrm{KL}(\pi||\pi_{k+1}^*)(s)]$. The motivation to do so is to get ride of the partition function. Yet, this is equivalent to what we call the "direct" approach, writing $Z_k \in \mathbb{R}^{\mathcal{S}}$ the partition function:

$$
\begin{aligned}
-\mathrm{KL}(\pi||\pi_{k+1}^*) &= \langle \pi, \ln \frac{\pi_k^\beta \exp \frac{\beta q_k}{\lambda}}{Z_k} - \ln \pi \rangle \\
&= \langle \pi, \ln(\pi_k^\beta \exp \frac{\beta q_k}{\lambda}) \rangle - \langle \pi, \ln Z_k \rangle - \langle \pi, \ln \pi \rangle \\
&= \frac{\beta}{\lambda} \langle \pi, q_k \rangle + \beta \langle \pi, \ln \pi_k \rangle - \langle \pi, \ln \pi \rangle - \ln Z_k \\
&= \frac{\beta}{\lambda} \left( \langle \pi, q_k \rangle + \lambda \langle \pi, \ln \pi_k \rangle - (\lambda + \tau)\langle \pi, \ln \pi \rangle - (\lambda + \tau) \ln Z_k \right) \\
&= \frac{\beta}{\lambda} \left( \langle \pi, q_k \rangle - \lambda \, \mathrm{KL}(\pi||\pi_k) + \tau \mathcal{H}(\pi) - \ln Z_k \right).
\end{aligned}
$$

So, up to the scaling $\frac{\beta}{\lambda} = \frac{1}{\lambda + \tau}$ and to the term $\ln Z_k$, which is a constant regarding the optimized policy $\pi$ and can thus safely be ignored, we obtain the loss of Eq. (31).

When considering DA-VI, the policy can be computed analytically, $\pi_{k+1} = \mathcal{G}^{0,\tau}(h_k)$, but $h_k$ has to be approximated (and can be seen as the logits of the policy). With $h_{k-1}$ and $q_k$ the target networks:

$$
\mathcal{L}_{\mathrm{da}}(h) = \hat{\mathbb{E}}_{s,a} \left[ ([\beta h_{k-1} + (1-\beta)q_k](s,a) - h(s,a))^2 \right].
$$

Minimizing this loss over networks $h$ gives $h_k$. This is reminiscent of momentum-DQN (see Appx. B.2).

**Evaluation step.** Given one of the three ways of doing the greedy step, one can choose between regularizing the evaluation step ($w/$, as suggested by the theory) or not ($w/o$, as often done empirically). This second case is already depicted in Eq. (30) (changing the considered policy) and the first case is given by

$$
\mathcal{L}_{w/}(q) = \hat{\mathbb{E}}_{s,a} \left[ \left( [\hat{T}_{\pi_{k+1}}^{\lambda,\tau} q_k](s,a) - q(s,a) \right)^2 \right].
$$

So combining one of the two evaluation steps ($w/$ or $w/o$) with one of the three greedy steps (MD-dir, MD-ind or DA), we get six variations. We discuss also the limit case without entropy.

**When $\tau = 0$.** For MD-VI, one can set $\tau = 0$. However, recall that for DA-VI, the resulting algorithm is different. DA-VI($\lambda$, 0) is not practical in a deep learning setting, as it requires averaging over iterations. Indeed, updates of target networks are too fast to consider them as new iterations, and a moving average is more convenient. Vieillard et al. [43] used a decay on $\beta$ to mimic this behavior, but this is a heuristic that needs to be tuned. Therefore, for DA-VI we will only consider the limit case $\lambda + \tau \to 0$ with $\beta = \frac{\lambda}{\lambda + \tau}$ kept constant (that is, momentum-DQN with fixed $\beta$). In this case, type 1 and 2 are equivalent. We offer additional visualisations in Appx. E.4.

### E.2 More on practical algorithms

We now detail the losses presented in the previous section, giving equations that are closer to implementation, and providing a detailed pseudo-code in Algorithm 2. Firts, let us introduce some notations. The $q$-value is represented by a neural network $Q_\theta$ of parameters $\theta$, and the policy is represented by a network $\Pi_\phi$ of parameters $\phi$. During training, the algorithms interact with an environment, and collect transitions $(s, a, r, s')$ that are stored in a FIFO replay buffer $\mathcal{B}$. The parameters of the networks are copied regularly into old versions of themselves, with target weights $\bar{\theta}$ and $\bar{\phi}$. The weights $\theta$ are optimized during the evaluation step, and $\phi$ during the greedy step.

### E.2.1   Evaluation step

All the actor-critics we consider have the same update rule of their critic – the $Q$-network. We consider two regressions targets, corresponding to regularizing the evaluation step or not. If not regularized, we define a regression target as

$$\hat{Q}_{w/o}(r, s') = r + \gamma \sum_{b \in \mathcal{A}} Q_{\bar{\theta}}(s', b)\Pi_{\phi}(b|s'),$$

and if regularized,

$$\hat{Q}_{w/}(r, s') = \hat{Q}_2(r, s') - \lambda \operatorname{KL}\left(\Pi_{\phi}\|\Pi_{\bar{\phi}}\right)(s') + \tau\mathcal{H}\left(\Pi_{\phi}\right)(s').$$

The weights $\theta$ are then updated by minimizing the following regression loss with a variant of SGD

$$\mathcal{L}_{w/-w/o}(\theta) = \hat{E}_{\mathcal{B}}\left[\left(Q_{\theta}(s, a) - \hat{Q}_{w/-w/o}(r, s')\right)^2\right]. \tag{32}$$

Note that if $\Pi_{\phi}$ was greedy with respect to $Q_{\bar{\theta}}$, using $\mathcal{L}_{w/o}$ would reduce to Deep $q$-networks (DQN) [27].

### E.2.2   Greedy step

Let us re-write in detail the three equations from Section E.1 that define three ways of performing the greedy step.

**MD-dir.**   The Direct MD update tackles directly the optimization problem derived from the greedy step. For convenience, we define a loss (the opposite of what we would like to maximize) that we minimize with SGD

$$\mathcal{L}_{\operatorname{dir}}(\phi) = \hat{E}_{\mathcal{B}}\left[-\sum_{b \in \mathcal{A}} Q_{\bar{\theta}}(s, b)\Pi_{\phi}(b|s) + \lambda \operatorname{KL}\left(\Pi_{\phi}\|\Pi_{\bar{\phi}}\right)(s') - \tau\mathcal{H}\left(\Pi_{\phi}\right)(s')\right]. \tag{33}$$

**MD-ind.**   The indirect version is based on the analytical result of the optimization problem corresponding to the greedy step. We show in Appendix B.1 that , at iteration $k$ of MD-VI$(\lambda, \tau)$, we have $\pi_{k+1} = \mathcal{G}_{\pi_k}^{\lambda, \tau}(q_k) \propto \pi_k^{\beta} \exp \frac{q_k}{\tau + \lambda}$. Hence, we would need to fit a target that approximates this maximizer, by defining $\hat{\Pi}(a|s)$ as

$$\hat{\Pi}(a|s) = \Pi_{\bar{\phi}}(a|s)^{\beta} \exp \frac{Q_{\bar{\theta}}(s, a)}{\lambda + \tau}\left(\sum_{b \in \mathcal{A}} \Pi_{\bar{\phi}}(b|s)^{\beta} \exp \frac{Q_{\bar{\theta}}(s, b)}{\lambda + \tau}\right)^{-1}.$$

However, the exponential term can cause numerical problems, so what we optimize during the evaluation step is actually the logarithm of the policy. To work around this, we define a network $L_{\phi}$ that represents the log-probabilities of a policy, and we define a regression target

$$\hat{L}(s, a) = \frac{\lambda L_{\bar{\phi}}(a|s) + Q_{\bar{\theta}}(s, a)}{\lambda + \tau} - \ln \sum_{b \in \mathcal{A}} \frac{\lambda L_{\bar{\phi}}(b|s) + Q_{\bar{\theta}}(s, b)}{\lambda + \tau},$$

and then we have $\hat{\Pi}(a|s) = \exp\left(\hat{L}(s, a)\right)$ and $\Pi_{\phi}(a|s) = \exp\left(L_{\phi}(a|s)\right)$. We then define a loss on the parameters $\phi$,

$$\mathcal{L}_{\operatorname{ind}}(\phi) = \hat{E}_{\mathcal{B}}\left[\operatorname{KL}\left(\hat{\Pi}\|\Pi_{\phi}\right)(s)\right]. \tag{34}$$

**DA.**   The dual averaging version is inspired by the DA-VI formulation. Instead of representing directly the policy, we estimate a moving average of the $q$-values, and then compute its softmax. The moving average is estimated via a network $H_{\phi}$, which fits a regression target

$$\hat{H}(s, a) = \beta H_{\bar{\phi}}(s, a) + (1 - \beta)Q_{\bar{\theta}}(s, a),$$

and the policy is defined as softmax over $H_\phi(s, \cdot)$,

$$\Pi_\phi(a|s) = \exp\frac{H_\phi(s,a)}{\tau} \left(\sum_b \exp\frac{H_\phi(s,b)}{\tau}\right)^{-1}.$$

The weights $\phi$ are optimized by minimizng the loss

$$\mathcal{L}_{da}(\phi) = \hat{E}_\mathcal{B}\left[\left(H_\phi(s,a) - \hat{H}(s,a)\right)^2\right]. \tag{35}$$

### E.2.3    Pseudo code

We give a general pseudo-code of the deep RL algorithms we used in Alg. 2. Notice that for a policy $\pi$, we define the $e$-greedy policy with respect to $\pi$ as the policy that takes a random action (uniformly on $\mathcal{A}$) with probability $e$, and follows $\pi$ with probability $1 - e$.

---

**Algorithm 2** (MD-dir | MD-ind | DA)

---

**Require:** $L_q(\theta)$ and $L_\pi(\phi)$, two losses, respectively for the evaluation and the greediness. The choice of these losses determines the algorithm, see Table 2.
**Require:** $K \in \mathbb{N}^*$ the number of steps, $C \in \mathbb{N}^*$ the update period, $F \in \mathbb{N}^*$ the interaction period.
**set** $\theta$, $\phi$ at random
**set** $Q_\theta$ the q-value network, $\Pi_\phi$ the policy network, as defined in Sec. E.2.
**set** $\mathcal{B} = \{\}$
**set** $\Pi_{\phi,e_k}$ the policy $e_k$-greedy w.r.t. $\Pi_\phi$
$\quad \bar{\theta} = \theta, \bar{\phi} = \phi$
$\quad$ **for** $1 \leq k \leq K$ **do**
$\quad\quad$ Collect a transition $t = (s, a, r, s')$ from $\Pi_{\phi,e_k}$
$\quad\quad \mathcal{B} \leftarrow \mathcal{B} \cup \{t\}$
$\quad\quad$ **if** $k \mod F == 0$ **then**
$\quad\quad\quad$ On a random batch of transitions $B_{q,k} \subset \mathcal{B}$, update $\theta$ with one step of SGD on $L_q$
$\quad\quad\quad$ On a random batch of transitions $B_{h,k} \subset \mathcal{B}$, update $\phi$ with one step of SGD on $L_\pi$
$\quad\quad$ **end if**
$\quad\quad$ **if** $k \mod C == 0$ **then**
$\quad\quad\quad \bar{\theta} \leftarrow \theta, \bar{\phi} \leftarrow \phi$
$\quad\quad$ **end if**
$\quad$ **end for**
**output** $\Pi_\phi$

---

Table 2: Resulting algorithms given the choice of losses in Algorithm 2

| | $L_\pi$ | | |
|---|---|---|---|
| $L_q$ | $\mathcal{L}_{dir}$ (Eq.(34)) | $\mathcal{L}_{ind}$ (Eq. (33)) | $\mathcal{L}_{da}$ (Eq. (35)) |
| $\mathcal{L}_{w/}$ (Eq. (32)) | MD-dir $w/$ | MD-ind $w/$ | DA $w/$ |
| $\mathcal{L}_{w/o}$ (Eq. (32)) | MD-dir $w/o$ | MD-ind $w/o$ | DA $w/o$ |

### E.3    Hyperparameters

We provide the hyperparameters used on the Atari environments in Table 3, and on the Gym environments in Table 4. We use the following notations to describe neural networks: FC $n$ is a fully connected layer with $n$ neurons; $\text{Conv}_{a,b}^d\, c$ is a 2d convolutional layer with $c$ filters of size $a \times b$ and a stride of $d$. All hyperparameters are the one found in the Dopamine code base. We only tuned the learning rate and the update period of DQN on Lunar Lander (not provided in Dopamine).

Table 3: Parameters used on Atari. Both the $Q$-network and policy-network have the same structure. $n_A$ is the number of actions available in a given game.

| Parameter | Value |
|---|---|
| $K$ (number of steps) | $5 * 10^7$ |
| $C$ (update period) | 8000 |
| $F$ (interaction period) | 4 |
| $\gamma$ (discount) | 0.99 |
| $|\mathcal{B}|$ (replay buffer size) | $10^6$ |
| $|B_{\pi,k}|$ and $|B_{q,k}|$ (batch size) | 32 |
| $e_k$ (random actions rate) | $e_0 = 0.01$, linear decay of period $2.5 \cdot 10^5$ steps |
| networks structure | $\mathrm{Conv}_{8,8}^4\, 32 - \mathrm{Conv}_{4,4}^2\, 64 - \mathrm{Conv}_{3,3}^1\, 64 - \mathrm{FC}\, 512 - \mathrm{FC}\, n_A$ |
| activations | Relu |
| optimizers | RMSprop ($lr = 0.00025$) |

Table 4: Parameters used on CartPole and Lunar Lander . Both the $Q$-network and policy-network have the same structure. We have $n_A = 2$ on CartPole, and $n_A = 8$ on Lunar Lander.

| Parameter | Value |
|---|---|
| $K$ (number of steps) | $5 * 10^5$ |
| $C$ (update period) | 100 (Cartpole), 2500 (Lunar Lander) |
| $F$ (interaction period) | 4 |
| $\gamma$ (discount) | 0.99 |
| $|\mathcal{B}|$ (replay buffer size) | $5 * 10^4$ |
| $|B_{\pi,k}|$ and $|B_{q,k}|$ (batch size) | 128 |
| $e_k$ (random actions rate) | 0.01 (constant with $k$) |
| networks structure | $\mathrm{FC}\, 512 - \mathrm{FC}\, 512 - \mathrm{FC}\, n_A$ |
| activations | Relu |
| optimizers | Adam ($lr = 0.001$) |

### E.4   Additional results

**Additional environment.**  In addition to the environments considered in Sec. 5, we provide three additional environments: Lunar Lander (from gym), Breakout and Seaquest (from Atari). The comments on these environments are similar to the discussion of Sec. 5

**Full tables.**  We also provide the full results of the experiments (those from Section 5 and the new ones). The same plots are reported, expect that we add the exact value of each grid cell for completeness. Results for Carpole and Lunarlander are provided in Figs. 4 and 5, while results for the considered Atari games (Asterix, Breakout and Seaquest) are reported in Figs. 6, 7 and 8.

**Training curves.**  We also report training curves on Atari. We report training curves of DA, MD-dir and MD-ind in Fig. 9 for Asterix, on Fig. 10 for Breakout, and on Fig. 11 for Seaquest. We report the training curves of the limit cases on these three games on Figs. 12, 13 and 14. In these figures, an *iteration* corresponds to 250000 training steps, and we report every iteration the undiscounted reward averaged over the last 100 episodes (the *averaged score*). The training curves are averaged over 3 random seeds.

The training curves give more hindsights on the performance of the algorithms. Indeed, the metric we used in the tables (the averaged score over all iteration) is partly flawed, because it could give a high score to an algorithm with a performance drop at the end of training. For example, the MD-dir method on Atari seems to benefit from regularizing the evaluation step (as unregularized evaluation suffers from a performance drop), which is less visible from the score tables. In almost all the cases, we do not observe such behaviour, which validates the use of our metric.

Figure 4: Cartpole with complete values.

Figure 5: Lunar Lander with complete values.

Figure 6: Asterix with complete values.

Figure 7: Breakout with complete values.

Figure 8: Seaquest with complete values.

Figure 9: All averaged training scores of MD-dir (top), MD-ind (middle) and DA (bottom), w/ and w/o, on Asterix, for several values of $\beta$ and $\tau$. Each plot corresponds to one value of $\beta$ (in the titles). In each plot, a curve corresponds to a value of $\tau$: $1e-3$ (orange), $3e-3$ (green), $1e-02$ (red), $3e-2$ (blue), $1e-1$ (brown). The blue dotted line is DQN.

Figure 10: All averaged training scores of MD-dir (top), MD-ind (middle) and DA (bottom), $w/$ and $w/o$, on Breakout, for several values of $\beta$ and $\tau$. Each plot corresponds to one value of $\beta$ (in the titles). In each plot, a curve corresponds to a value of $\tau$: $1e-3$ (orange), $3e-3$ (green), $1e-02$ (red), $3e-2$ (blue), $1e-1$ (brown). The blue dotted line is DQN.

Figure 11: All averaged training scores of MD-dir (top), MD-ind (middle) and DA (bottom), $w/$ and $w/o$, on Seaquest, for several values of $\beta$ and $\tau$. Each plot corresponds to one value of $\beta$ (in the titles). In each plot, a curve corresponds to a value of $\tau$: $1e-3$ (orange), $3e-3$ (green), $1e-02$ (red), $3e-2$ (blue), $1e-1$ (brown). The blue dotted line is DQN.

Figure 12: All averaged training scores of limit cases on Asterix, for several values of $\beta$ and $\lambda$. In each plot, a curve corresponds to a value of $\lambda$ for MD-ind and MD-dir, and to a value of $\beta$ for Mo-DQN. The blue dotted line is DQN.

Figure 13: All averaged training scores of limit cases on Breakout, for several values of $\beta$ and $\lambda$. In each plot, a curve corresponds to a value of $\lambda$ for MD-ind and MD-dir, and to a value of $\beta$ for Mo-DQN. The blue dotted line is DQN.

Figure 14: All averaged training scores of limit cases on Seaquest, for several values of $\beta$ and $\lambda$. In each plot, a curve corresponds to a value of $\lambda$ for MD-ind and MD-dir, and to a value of $\beta$ for Mo-DQN. The blue dotted line is DQN.