[Reviews · NeurIPS 2020]

Review 1

Summary and Contributions: This paper analyzes the effects of KL and entropy regularization in RL. It unifies many existing regularized RL algorithms under one framework and proves strong theoretical results for approximate value iteration with KL regularization that improves dependence on horizon and error over prior work. It then conducts a careful analysis both theoretical and empirical of how KL regularization interacts with entropy regularization.

Strengths: 1. All the main theoretical and empirical claims seem to be sound, and I think the paper provides useful insight to the community on regularization in RL. 2. The paper provides a nice unifying framework to compare prior methods and illustrates a useful equivalence between the mirror descent and dual averaging variants of MPI. 3. The main theorem provides tighter bounds on a KL-regularized variant of approximate value iteration. These improve on bounds from prior work in the dependence on horizon (from quadratic to linear) and errors (norm of sum vs. sum of norms) while having a slower asymptotic convergence without errors (1/k vs. gamma^k). 4. The paper provides clear comparisons of the rates in each term of each theorem with prior work and provides nice intuition about how each term related the constants defining the problem to their practical consequences. 5. The paper clearly enumerates the limitations of the scope of the theoretical analysis (no benefit to entropy alone, no statement about how to control errors, assume ability to perform exact greedy step). This is very useful to situate the claims of the paper in the broader context of the full RL problem. 6. The empirical evaluation covers some of the limits of the theory, e.g. going to neural networks in problems without full access to the MDP or ability to exactly compute the greedy step. Multiple environments with large sweeps are tested, and the results are clearly presented.

Weaknesses: I don't see any major weaknesses with this paper. One minor weakness is that the experiments could cover the gap between the theory and practice a little more comprehensively. I understand that to recover most deep RL algorithms an actor must be added and then the greedy step becomes approximate. But with a finite action space (or even low-dimensional continuous like cartpole where the action space can be discretized) it should be possible to compute the regularized greedy policy exactly at any state from the Q-values for all actions from that state as explained in appendix A. Evaluating variants like this would seem to be important to isolate which part of the approximation (i.e. the evaluation or the greedy step) causes the observed effects.

Correctness: Yes, I think that the claims and method are correct.

Clarity: 1. The paper is generally clear. I especially appreciated the clear interpretations of the theorems, enumeration of limitations of the analysis, and well structured experiment section with clear goals and conclusions. 2. One point of confusion is switching between referring to value iteration and referring to policy iteration. My understanding (after a bit of confusion) is that the case of m=1 approximate value iteration and general m is approximate modified policy iteration, this should be stated and emphasized more clearly. It also seems that all of the theory and experiments use m=1, so it is not clear if this fully general setup is necessary in the main text (although it is useful in appendix B).

Relation to Prior Work: 1. The connection to prior work is clear throughout. The unifying analysis of how prior algorithms fit into the AMPI framework is helpful. 2. Bounds in Theorem 1 are better than prior work in combining a linear dependence on horizon with the norm of a sum of errors (rather than sum of norms). 3. Theorem 2 is not clearly original, but proof is different and analysis connecting to the broader story of the interaction of KL and entropy is helpful.

Reproducibility: Yes

Additional Feedback: The font is not the standard neurips font (and actually seems bigger than neurips font, so I don't think this violates the page limit). The spacing between lines also seems to be non-standard. Typos and minor unclear things: 1. lines 43 and 48 use "if" to present the results as hypotheticals, which is confusing since these are the claims of the paper 2. it would be helpful to restate what beta is before stating Theorem 2 3. line 237 should be "errors propagate" 4. line 263 should be "that we will call" 5. line 316 should be "MD-indirect provides the best result" based on my reading of the plot 6. line 484 should be "Bellman"


Review 2

Summary and Contributions: The paper provides a new theoretical framework for investigating why KL regularisation is effective in RL. Many existing algorithms that use entropy/KL regularisation fit into a mirror descent modified policy iteration scheme (MD-MPI). The authors introduce a new theoretical framework is based on a dual averaging formulation of modified policy iteration (DA-MPI), which can be shown to be equivalent to MD-MPI in most cases. The DA-MPI framework allows for the derivation of a novel performance bound for regularised objectives, which explains the benefits of using regularisation in terms of a linear dependency on the horizon and an averaging effect of the estimation error. The derivation of the bound only applies if KL or KL and entropy regularisation is used, and not solely entropy regularisation. Moreover, it assumes that the policy improvement step can be carried out exactly which is impractical when nonlinear value and policy function approximators are used or action spaces are not small and discrete. To account for this, an empirical evaluation is carried out across two discrete MDPs, confirming that the theoretical results carry over when the assumptions underpinning them aren't met.

Strengths: The paper offers two novel theoretical contributions. The first contribution is the introduction of the novel DA-MPI framework, which the authors prove is equivalent to the MD-MPI scheme (proposition 1). The second contribution is the derivation of two new performance bounds based on this framework, one for a purely KL-regularised objective and another for a KL and entropy regularised objective (theorems 1 and 2). These bounds have been missing from existing analysis and offer an important contribution to the NeurIPS community. The insight offered in the analysis of these bounds is valuable: we see that the first bound demonstrates that using KL-regularisation leads to a linear dependence on the horizon term, unlike the typical quadratic dependence for non regularised form. Secondly, the error term is a an average of errors, rather than a sum of norms. This means that should the errors form a martingale, the MCT proves that the eventual error term will tend to zero and hence the policy converges to the optimal policy. Conversely, a sum of norms can never converge. The second bound derived sheds light on the effect of introducing an entropy term. Whilst the error term in this instance is a sum of norms, the resulting bound has a faster asymptotic rate of convergence. In light of these insights, the authors offer a discussion of how to trade of the entropy and kl terms in the objective which will be indispensable in guiding those in the community wishing to implement objectives that involve regularisation The authors then make an empirical evaluation for cases where the theoretical assumptions do not hold. This provides evidence for four key hypothesise, that is: KL regularization is beneficial, regularizing the evaluation step helps, the best choice of greediness is problem dependent and entropy is better with KL, and KL alone might be sufficient. These results provide a reassuring confirmation and summary of the fundamental principles of entropy and KL regularisation in RL from a non-probabilistic perspective.

Weaknesses: The most obvious criticism of this paper is the relatively limited number of domains for the empirical evaluation. The strength of the empirical claims in Section 5 would be greatly improved if they where shown to hold across a wider range of environments. Moreover, algorithms such as SAC are tailored to continuous control domains, and so evaluation in these domains would be useful to see. Another weakeness is that many of the KL and entropy regularisation techniques find motivation from a probabilistic perspective of the control problem [1]. This perspective is not acknowledged or discussed by the authors. Indeed, optimality of the policy and convergence under an EM style algorithm given similar assumptions of a complete greedification step has had theoretical treatment before in certain contexts with KL regularisation (interpreted as a policy prior) and entropy regularisation (interpreted as a uniform prior) under this probabilistic perspective [2], although unlike in this work, bounds are not derived. I found two typos: one on line 263, 'cal' and one on line 299, 'ony' [1] Reinforcement Learning and Control as Probabilistic Inference: Tutorial and Review, Levine 2018 [2] VIREL: A variational inference framework for reinforcement learning. Fellows 2019

Correctness: I have found no issues in a reading of the proofs for proposition 1 and theorems 1 and 2. I will derive these results with a pen and paper later given enough time.

Clarity: Overall, the paper is exceptionally well written. It clearly conveys the abstract ideas underpinning the theory, provides rich discussion and insight at all points and leaves the reader with clear take home messages. The structuring of the paper is logical and helps understanding. Sketches of proofs are always given first, which makes tackling the full proof more accessible. I also appreciated the detailed discussions in the appendix, which supplemented the higher level presentation of material in the main text. Natural questions that the reader would ask when reading the paper are addressed and the authors are honest and thorough in discussing the limitations of their theory. My only criticism is that the results in Fig. 2 are difficult to read. Making the figure and annotations bigger would greatly improve the exposition of Section 5. There's a lot of blank space on page 6, so this wouldn't require much effort to achieve. From a purely aesthetic perspective, it's also annoying that the two experiments left and right in Fig. 2 don't align and that the scale of the bars for the limiting cases vary.

Relation to Prior Work: The authors provide a clear tabular overview of how their framework encompasses existing algorithms and contrast the bounds derived from their theoretical results with existing bounds without regularisation. This is further supplemented by Section B in the Appendix which clearly derives many existing algorithms from the theoretical framework.

Reproducibility: Yes

Additional Feedback: I appreciated the rebuttal from the authors over my concerns raised over including relevant RL as inference references. I look forward to reading the updated paper, which I maintain is a solid contribution.


Review 3

Summary and Contributions: The paper has a deep look at empirical value iteration/policy iteration with entropy+KL regularization. There are two main contributions: first that these algorithms are roughly equivalent to many other recent algorithms, and the main contribution is a tight error bound (linear dependence on the virtual horizon for the MDP).

Strengths: It is interesting to see all the links with other algorithms, and the two main results with error bounds provides a great deal of intuition for the relative value of KL regularization. The empirical study helps to reinforce this intuition.

Weaknesses: Proposition 1 by itself does not seem to be very deep (unless I am missing something), but the conclusions regarding other algorithms is remarkable -- the authors seem to believe every famous algorithm is covered! However, the discussion after Proposition 1 is too loose. It isn't at all clear how there could be a connection with advantage learning. It isn't at all convincing that this sort of "blind regularization" (agnostic to reward) will be the future of RL. But I bet the ideas here could extend to reward-oriented regularization. The empirical study is elegant, but a bit hard to follow. I hope they can devote a bit more space to explaining the data, and a bit less open ended discussion earlier.

Correctness: I did not go through the proofs in detail, but the authors make a convincing case. The simulation study is a great bonus.

Clarity: Very good. The analysis of an abstraction of RL is a blessing to help clarify the exposition. The summary of prior work is great, given the limited space.

Relation to Prior Work: Two papers related in terms of error bounds dependent on the horizon: Paper 1 has mean-square-error bound independent of horizon, by learning the "relative Q function" (the Q function minus a constant). Paper 2 establishes a bound that grows linearly with horizon (this second reference is probably contemporary with the paper under review; both papers are on arXiv). 1. Q-learning with Uniformly Bounded Variance: Large Discounting is Not a Barrier to Fast Learning Adithya M. Devraj, Sean P. Meyn [Submitted on 24 Feb 2020 (v1), last revised 7 Jul 2020 (this version, v2)] 2. Almost Optimal Model-Free Reinforcement Learning via Reference-Advantage Decomposition Zihan Zhang, Yuan Zhou, Xiangyang Ji [Submitted on 21 Apr 2020 (v1), last revised 6 Jun 2020 (this version, v2)]

Reproducibility: Yes

Additional Feedback:


Review 4

Summary and Contributions: To enforce entropy regularization term has been frequently adopted to encourage exploration and accelerate policy optimization in reinforcement learning. The authors carry out both theoretical and empirical analysis on the use of KL regularization showing that it implicitly averages q-values. They further provide performance bounds with discussions on the convergence rate, horizon term and error term respectively. Some empirical results are presented to aid the theoretical analysis.

Strengths: The theoretical analysis and detailed discussions on the error propagation with KL regularization could potentially provide insights into algorithm design.

Weaknesses: Given the thoroughness of the analysis, only existing algorithms are studied and the novelty is limited by design. Also even though in the analysis a bunch of existing algorithms are included and their connection are discussed, the discussion could be improved by being more clear about the nature of these connections for example what do algorithms that consider the same regularization share in common.

Correctness: The technical details make sense to me.

Clarity: The paper is well written and organized, with minor issue: at Line 56 the parameter \tau is used without definition.

Relation to Prior Work: There are enough discussions on prior work.

Reproducibility: Yes

Additional Feedback:

[Author Response · NeurIPS 2020]

First, we would like to thank the reviewers for the time spent on assessing our work, for their positive feedback, and for their useful comments and suggestions.

**Reviewer #1**

**With a finite action space [...] it should be possible to compute the regularized greedy policy exactly.** With a nonlinear parameterization, we do not think it to be possible (the greedy policy depends on the Q-values but also on the previous policy), except if one can afford to remember all past Q-values (see the Eq. l.94).
**About $m$.** Yes, $m = 1$ corresponds to VI, and the general scheme to MPI. We'll clarify further. Even if the analysis only holds for $m = 1$ (and its extension to $m > 1$ is not obvious), we think important to provide the abstract scheme for the general case, to cover a wider range of existing algorithms and to ease the connections. Also, we think interesting that the equivalence (stated in Prop. 1) holds in the general case.
**Font.** We double-checked, and we use the provided Neurips style file. We'll triple-check.
**Typos.** Thank you, we'll correct them.

**Reviewer #2**

**Limited number of domains.** We consider only two domains in the empirical part of the main paper, due to the page limit. However, the reviewer has maybe missed the additional domains we provide in Appx. E.4. In total, we evaluate our algorithms on (1) 100 garnets (random MDPs), (2) 2 gym environments: CartPole and LunarLander, and (3) 3 Atari games: Asterix, Breakout and Seaquest. More would be better, but our finding are consistent across all these domains.
**Control as probabilistic inference.** We acknowledge that there are connections between entropy/KL-regularization and control as probabilistic inference. The formalism we adopt is rather the one of [19], where the link with probabilistic inference is discussed. That's true that it should be discussed here too, so we'll add a discussion about this in the final version.
**Figure 2.** Thank you, we will fix that. Notice also that these figures are provided bigger in the appendix (as well as additional visualizations).

**Reviewer #3**

**Discussion after Prop 1 is too loose.** This discussion is indeed quite dense (page limit), but it is developed at length in the whole Appx. B.
**Connection to AL.** We confirm this connection, it is explained in Appx. B.2, l.566-569. If not clear enough here, we will expand the explanation. Shortly, CVI is a reparameterization of MD-MVI (as shown in Appx B.2), and AL is a limiting case of CVI (as the temperature goes to zero), hence the connection.
**Related papers.** Thank you, we were not aware of these papers. We will make sure to discuss them in the final version.

**Reviewer #5**

**The discussion could be improved by being more clear about the nature of these connections.** The discussion in Sec. 3 is indeed dense, but it is expanded at length in the whole Appx. B.
$\tau$ **is used without definition.** We write "For $\tau \geq 0$" as a short way for "For any real number $\tau \geq 0$". We precise that it is a temperature later, when relevant.

[Meta-Review · NeurIPS 2020]

This paper unifies KL and entropy regularized RL algorithms (e.g. TRPO, MPO, SAC, soft Q-learning, softmax DQN, DPP, etc) under one mirror descent framework and provides proofs for KL regularized value iteration. The paper shows that using KL regularization implicitly averages the estimates of the Q function, and using this result it shows a linear dependence of the approximation error of Q on the time horizon, whereas in many previous works with similar assumptions it was quadratic. This is a significant result. In addition, KL regularization ensures convergence in the case of independent and centered errors, which is not the case for standard approximate dynamic programming. The paper also examines how KL regularization interacts with entropy regularization, and presents empirical findings suggesting that KL regularization alone might be sufficient and better then entropy regularization, encouraging a lot of exploration in the beginning and less so as the policy deviates from being uniform. This is the highest-rated paper in my batch. All reviews were very positive, and reviewers unanimously thought this was a very strong contribution, Most criticisms were cosmetic in nature and easily fixable. R1 had concerns about a larger range of experiments, but it was pointed out in the rebuttal that they exist in the appendix. The contributions are impactful and the work will be of interest to not only the RL theory community but also users of RL methods. I recommend this paper be accepted as an oral, both because of its merits and to reach as broad an audience as possible.